



# EUNADICS early warning system dedicated to support aviation in case of crisis from natural airborne hazard and radionuclide cloud.

Hugues Brenot[1], Nicolas Theys[1], Lieven Clarisse[2], Jeroen van Gent[1], Daniel R. Hurtmans[2], Sophie Vandenbussche[1], Nikolaos Papagiannopoulos[3], Lucia Mona[3], Timo Virtanen[4], Andreas Uppstu[4], Mikhail Sofiev[4], Luca Bugliaro[5], Margarita Vázquez-Navarro[5,*], Pascal Hedelt[5], Michelle Maree Parks[6], Sara Barsotti[6], Mauro Coltelli[7], William Moreland[7**], Delia Arnold-Arias[8,9], Marcus Hirtl[8], Tuomas Peltonen[10], Juhani Lahtinen[10], Klaus Sievers[11], Florian Lipok[12], Rolf Rüfenacht[13], Alexander Haefele[13], Maxime Hervo[13], Saskia Wagenaar[14], Wim Som de Cerff[14], Jos de Laat[14], Arnoud Apituley[14], Piet Stammes[14], Quentin Laffineur[15], Andy Delcloo[15], Robertson Lennart[16], Carl-Herbert Rokitansky[17], Arturo Vargas[18], Markus Kerschbaum[19], Christian Resch[20], Raimund Zopp[21], Matthieu Plu[22], Vincent-Henri Peuch[23], Michel Van Roozendael[1], and Gerhard Wotawa[8]

[1]Royal Belgian Institute for Space Aeronomy (BIRA), Brussels, 1180, Belgium
[2]Service Spectroscopy, Quantum Chemistry and Atmospheric Remote Sensing (SQUARES), Université Libre de Bruxelles (ULB), Brussels, 1050, Belgium
[3]Consiglio Nazionale delle Ricerche, Istituto di Metodologie per l'Analisi Ambientale (CNR-IMAA), Tito Scalo (PZ), 85050, Italy
[4]Finnish Meteorological Institute (FMI), Helsinski, 00101, Finland
[5]German Aerospace Center (DLR), Oberpfaffenhofen, Germany
[6]Icelandic Meteorological Office (IMO), Reykjavík, 105, Iceland
[7]Osservatorio Etneo, Istituto Nazionale di Geofisica e Vulcanologia, Catania, 95125, Italy
[8]Zentralanstalt für Meteorologie und Geodynamik (ZAMG), Vienna, 1190, Austria
[9]Arnold Scientific Consulting, Manresa, 08242, Spain
[10]Radiation and Nuclear Safety Authority (STUK), Helsinki, 00880 , Finland
[11]Klaus Sievers Aviation Weather (KSAW), Lenggries, 83661, Germany
[12]Bridging Markets and Technologies Services Gmbh (BRIMATECH), Vienna, 1030, Austria
[13] Federal Office of Meteorology and Climatology MeteoSwiss, Payerne, 1530, Switzerland
[14]Royal Netherlands Meteorological Institute (KNMI), De Bilt, 3731 GK, the Netherlands
[15]Royal Meteorological Institute of Belgium (KMI-IRM), Brussels, 1180, Belgium
[16]Swedish Meteorological and Hydrological Institute (SMHI), Norrköping, 601 76, Sweden
[17]Paris-Lodron-University Salzburg (PLUS), 5020, Austria
[18]Institute of Energy Technologies, Universitat Politecnica de Catalunya (UPC), Barcelona, 08028, Spain
[19]Austro Control Oesterreichische Gesellschaft für Zivilluftfahrt Mbh (ACG), Schwechat, 1300, Austria
[20]Bundesministerium für Landesverteidigung und Sport (BMLVS), Vienna, 1090, Austria
[21]Flightkeys (FLIGHTKEYS), Vienna, 1070, Austria
[22]National Centre for Meteorological Research (CNRM / Météo-France), Toulouse, 31057, France
[23]European Centre for Medium-Range Weather Forecasts (ECMWF), Reading, RG2 9AX, United Kingdom
[*]now at EUMETSAT
[**]now at the University of Iceland

*Correspondence to*: Hugues Brenot (brenot@aeronomie.be)

**Abstract.**

The purpose of the EUNADICS prototype Early Warning System (EWS) is to proceed the combined use of harmonised data products from satellite, ground-based and in situ instruments to produce alerts of airborne hazard (volcanic, dust, smoke and radionuclide clouds), satisfying the requirement of ATM stakeholders (www.eunadics.eu). The alert products developed by EUNADICS EWS (i.e. NRT observations, email notifications and NetCDF Alert data Products, called NCAP) have shown shows the significant interest in using




selective detection of natural airborne hazards from polar orbiting satellite. The combination of several sensors inside a single global system demonstrates the advantage of using a triggered approach to obtain selective

detection from observations, which cannot initially discriminate the different aerosol types. Satellite products from hyperspectral UV and IR sensors (e.g. TROPOMI, IASI) and broadband geostationary imager (SEVIRI), and retrievals from ground-based networks (e.g. EARLINET, E-PROFILE and the regional network from volcanic observatories), are combined by our system to create tailored alert products (e.g. selective ash detection, $SO_2$ column and plume height, dust cloud and smoke from wildfires). A total of 23 different alert products are

implemented, using 1 geostationary and 13 polar orbiting satellite platforms, 3 external existing service, 2 EU and 2 regional ground-based networks. This allows the identification and the tracking of extreme events. EUNADICS EWS has also shown the interest to proceed a future relay of radiological data (gamma dose rate and radionuclides concentrations in ground-level air) in case of nuclear accident, highlighting the capability of operating early warnings with the use of homogenised dataset. For the four types of airborne hazard, EUNADICS EWS has

demonstrated its capability to provide NRT alert data products to trigger data assimilation and dispersion modelling providing forecasts, and inverse modelling for source term estimate. All our alert data products (NCAP files) are not publicly disseminated. Access to our alert products is currently restricted to key users (i.e. Volcanic Ash Advisory Centres, National Meteorological Services, World Meteorological Organization, governments, volcanic observatories and research collaborators), as these are considered pre-decisional products. On the other

hand, thanks to the SACS/EUNADICS web interface (https://sacs.aeronomie.be), the main part of the satellite observations used by EUNADICS EWS, are shown in NRT, with public email notification of volcanic emission and delivery of tailored images and NCAP files. All the ATM stakeholders (e.g. pilots, airlines and passengers) can access and benefit of these alert products through this free channel.

## 1 Introduction

Hazardous clouds can very often conduct to a considerable threat to human society, especially for the life, health and properties of population. A such threat can come from gas and aerosols emissions in the vicinity of a volcano (Baxter et al., 1999; Forbes et al., 2003; Hansell et al., 2006), in the surrounding area of a desert (Tobias et al., 2006), nearby a wildfire (Fowler, 2003), or due to a nuclear accident (Bennett et al., 2006). Due to atmospheric transport, airborne particle cloud may also impact area several thousand kilometres far from the source, with

implication in meteorological processing (Knippertz and Todd, 2012), causing worrying implication for the aviation (Casadevall, 1994; Casadevall et al., 1996; Miller and Casadevall, 1999; Guffanti et al., 2010; Tulet and Villeneuve, 2011; AlKheder and AlKandari, 2020; Khaykin et al, 2020) and critical deposition (Panebianco et al., 2017; Easdale and Bruzzone, 2018; Ridley et al., 2012; Zheng et al., 2020; Smith and Clark, 1986; Persson et al., 1987). The transport of hazardous gas and aerosols cloud (e.g. $SO_2$, ash, dust, smoke, radionuclide) is a challenge

for modellers, especially when a mix of particle occurs (Koch et al., 2006; Evangeliou and Eckhardt, 2020). The use of satellite (Prata, 2009, Prata et al., 2010; Theys et al., 2013, 2019; Clarisse et al., 2013, 2020; Christian et al., 2020; Khaykin et al, 2020) and ground-based networks (Ansmann et al., 2011; Pappalardo et al., 2013; D'amico et al. 2015; Osborne et al., 2019; Ansmann et al., 2020; Hernández-Ceballos et al., 2020) is an essential piece in the dispersion modelling process. It makes it possible as it can provide information about the source of

emission, discriminate the type of particles, and provide geolocation of the hazardous cloud, a crucial input for transport models.




### 1.1 Motivation

The infrastructures of the international operational meteorological communities play a critical role in the effort to strengthen disaster resilience. To address this particular responsibility in Europe, 31 members of the network of

European National Meteorological Services (NMSs) have signed an agreement for the establishment of the economic interest grouping, defined as the EIG EUMETNET (www.eumetnet.eu), with the aim of developing this proposal. It is well recognised that hydrometeorological events constitute the large majority of all disasters that occur worldwide (WMO, 2020). In this area, a lot has been achieved under the EUMETNET umbrella, for example the establishment of the METEOALARM program (www.meteoalarm.eu). Additional work, however, is needed

with regard to a hazard category that we refer to as "airborne hazards" (environmental emergency scenarios), including volcano eruptions, nuclear accidents, forest fires and desert dust events. Therefore, the logical next step is the expansion of EUMETNET activities into the emergency response coordination area. In terms of observations, EUMETNET has expanded its E-PROFILE program with a dense network of automatic lidars and ceilometers providing qualitative aerosol information (ash, dust, smoke and pollution) in real time

(www.eumetnet.eu/e-profile). In 2020, EUMETNET approved a business case for the further development of the E-PROFILE lidar network which foresees the implementation of a processing chain for quantitative aerosol mass estimates by 2023. This service will increase EUMETNET's capabilities with regards to airborne hazards.

Aviation is one of the most critical infrastructures of the 21$^{st}$ century. Even comparably short interruptions can cause economic damage summing up to the Billion-Euro range (IATA, 2010). As evident from the past, aviation

shows certain vulnerability with regard to natural hazards. Safe flight operations, air traffic management and air traffic control is a shared responsibility of EUROCONTROL, national authorities, airlines and pilots. All stakeholders have one common goal, namely to warrant and maintain the safety of flight crews and passengers. Currently, however, there is a significant gap in the Europe-wide availability of real time hazard measurement and monitoring information for airborne hazards describing "what, where, how much" in 3 dimensions, combined

with a near real-time (NRT) European data analysis and assimilation system. In practice, this gap creates circumstances where various stakeholders in the system may base their decisions on different data and information.

### 1.2 Overview of EUNADICS project

This work has been conducted within the framework of the EUNADICS-AV project, which received funding from

the European Union's Horizon 2020 research programme (https://ec.europa.eu/programmes/horizon2020). This European H2020 project was launched in October 2016. The projects has received funding for 3 years and was completed in September 2019 (www.eunadics.eu). The acronym EUNADICS-AV is for European Natural Airborne Disaster Information and Coordination System for Aviation. We will short it by EUNADICS in the following text of this study.

The main objective of EUNADICS is to close this gap in data and information availability, enabling all stakeholders in the aviation system to obtain fast, coherent and consistent information. This would facilitate the work of all stakeholders in the system, on one hand the European Aviation Crisis Coordination Cell (EACCC), the Air Traffic Management (ATM) and Air Traffic Control (ATC) functions, and on the other hand airline flight dispatching and individual flight planning. The idea of this prototype mechanism is to take into account and create

input to existing national and international systems, including the Volcanic Ash Advisory Centres (VAACs)



delivering products for aviation in case of a volcanic eruption, and the Wolrd Meteorological Organization (WMO) designated Regional Specialised Meteorological Centres (RSMCs) with activity specialisation Atmospheric Transport Modelling in charge of providing products in case of a nuclear accident and emergency. The objective of EUNADICS is driven by the strategic target as formulated in the Communication from the European Commission to the Institutions (COM 670, 2011), setting up an Aviation Safety Management System for Europe, i.e. by improving the quality of safety information, by sharing the information and the results of analysis, and by reaching agreement on those risks where coordinated action will bring the greatest benefits.

The technical objectives of EUNADICS are the following:

- to facilitate coherent Pan-European risk and exposure assessments for aviation related to airborne hazards, and to collect and consolidate requirements from the various stakeholders. Airborne hazards would include (i) volcanic ash/$SO_2$ dispersion, (ii) nuclear emissions, (iii) forest fires and (iv) sand storms.
- to integrate and harmonise existing observing systems and infrastructures in Europe, in particular aerosols, trace gases and radioactivity
- to improve the quality of data and related analysis products available in an emergency situation, most importantly by integrating vertical profile information into the data assimilation and analysis
- to assure pan-European information accessibility for aviation stakeholders in a crisis situation by providing an interoperable pilot implementation of the EUNADICS data and information system
- and to validate/test such a system in a realistic framework

EUNADICS system includes hazards from natural events, technical accidents as well as wilful acts, as long as these hazards have at least a mesoscale impact (>10 km).

The results of EUNADICS have been achieved according to the following basic principles:

- It is based on existing infrastructures, assignments and assets in Europe, including the International Civil Aviation Organization (ICAO)/WMO VAAC and WMO RSMC system.
- A complementarity work has been established with other related EU projects and initiatives, in particular with the Copernicus Atmosphere Service — CAMS (https://atmosphere.copernicus.eu) and EURDEP (https://remon.jrc.ec.europa.eu), and other international services.
- As the project will report to EUMETNET policy making organs like the Science and Technology Advisory Committee (STAC) and subsequently the Assembly, a direct horizontal link and coordination mechanism on a technical level across European governments is assured.
- NMS involvement assure a direct and pragmatic link to crisis responders, authorities, airlines, pilots and citizens, which has been exploited during the project.
- Direct involvement of a service provider for flight scheduling, an Air Navigation Service Provider (ANSP) and the military in one EU member state (Austria) allowed to address issues of civil-military coordination with regard to airborne hazards.

It should also be mentioned here that it was not an objective of EUNADICS to put in place a new pan-European forecast system for environmental hazards. Forecast systems do exist at national level as well as under international frameworks and responsibilities, for example operated by WMO RSMCs with activity specialisation



atmospheric transport modelling and by the VAACs, as described by Lechner et al. (2018). These Centres, as well as NMSs, are users of our system, which assures that all national and international downstream users continue getting their products through established and tested channels.

The EUNADICS consortium consists of 21 participating organisations from 12 different countries. It includes

National Meteorological Services, Monitoring data providers, Operational Volcanologists, Small and Medium-sized Entreprises, a University institute, an Air Navigation Service provider and a Military Organisation.

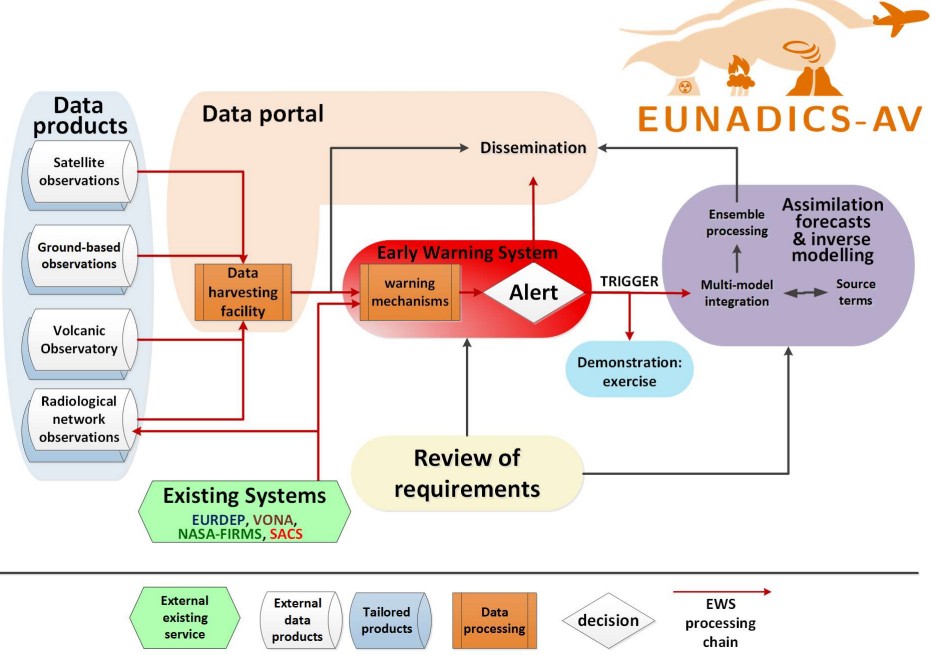

Figure 1: Overview of EUNADICS activities with a separation in blocs.

The activities of EUNADICS are illustrated in the diagram of Fig. 1, with the following methodologies (blocs in

brackets):

- Stakeholders requirements, hazard risk assessments and international cooperation (Review of requirements)
- Measurement from ground-based networks, satellites data products, observations from observatories (Data products)

- Data harvesting facility and dissemination (Data portal)
- Warning mechanisms and alert product development (Early Warning System)
- Data Dissemination – EUNADICS-AV portal
- Multi-model integration, source terms, ensemble modelling (Assimilation, forecasts and inverse modelling)

- Aviation product development, system testing and exercises (Demonstration exercise)

EUNADICS project took place under the supervision of an advisory and users board, which included a representative from EUMETNET, CAMS, EUROCONTROL, ESA, WMO, EASA (European Union Aviation Safety Agency), and from the European Cockpit Association.



### 1.3 Role of this work in EUNADICS

This study focusses on EUNADICS Early Warning System (EWS). This bloc/activity has a central role in EUNADICS system as it acts as a trigger for the data integration of tailored observations in dispersion model forecasts, which can provide critical information in the resilience process for ATM decision-makers facing a crisis due to airborne hazard. As shown by the processing chain in Fig. 1, the EWS depends on inputs of observations, carried out by the data harvesting facility that acts as a primordial phase of the hosting platform of EUNADICS

EWS. After a presentation of the requirements (i.e. users requirements, observational data inventories, review of the input for the data integration and of the external reports), this study presents the concept, mechanism of EUNADICS EWS, with details about data products from satellite, ground-based and in situ platforms used to produce alerts. The pre-alerting mechanism and data provision is then described (with service description, performance verification, and cases studies). Finally, conclusion and future developments are presented.

### 2 Requirements for EUNADICS Early Warning System

After an overview of requirement reports (from users, data inventories, data integration and other sources) which cover the activity blocs addressed by EUNADICS, a summary of the requirements tackled by our EWS is presented.

### 2.1 Review of user requirements

**Table 1: Overview of user requirements.**

| 1. Quality of information | a) optimisation of the risk assessment/re-routing/response action for airline/aircraft operators by using | | b) improvement of the information reliability by using quality labels for single scenario forecasts providing |
|---|---|---|---|
| | i. | improved NRT observations (better time-space 2D/3D resolution) | i. level of confidence<br>ii. errors bars |
| | ii. | charts of tailored products (synergy between different kind of observations) | |
| | iii. | better particle characterisation (sort, size) | |
| | iv. | improved forecast charts (better time-space 2D/3D resolution) | |
| | v. | multiple scenario forecasts and probabilistic ensemble | |
| | vi. | upgrade forecasts based on new information (not relying on systematic time-line) | |
| | vii. | forecasts of a fake natural hazard or radioactive plume to improve the planning of airline companies (exercise) | |
| 2. System interoperability | a) | automated data integration via a portal | |
| | b) | System-Wide Information Management (SWIM)-compliant information (readable by all airspace users: i.e. software used by airlines, weather charts, SIGnificant METeorological Information messages — SIGMETs) | |
| | c) | bundling of all available information (with quality labels) | |
| | d) | restricted access for key users (e.g. EUROCONTROL, EASA, VAACs) to avoid misunderstanding of public users | |
| 3. Improvement of the display (requirements from pilots and airlines) | a) | use of hazard icon in charts | |
| | b) | visualisation of hazard dispersion (obs. and forecast charts) using aviation stakeholders maps | |
| | c) | use of concentrations at flight levels | |
| | d) | use of specific maps for pilots with information related to their own flight only | |
| | e) | combination of satellite images with maps used in aviation | |





A substantial report of user requirements has been established by EUNADICS partners. The main parties consulted was the VAACs, EUROCONTROL, ICAO, WMO, EASA, the airline companies, governmental institutions and primarily, the pilots and passengers. Table 1 presents a highlight of the three types of requirements identified from this report, i.e. Quality of information, System interoperability and Improvement of display.

### 2.2 Review of observational data inventories

Reviews of data inventories of satellite, ground-based and in situ products have been used to determine the most relevant products to be implemented in our EWS, according to user requirements presented in the previous section. We identified two types of products, i.e. existing NRT (or proven NRT) products and tailored products in development. Key products, allowing NRT visualisation of natural airborne hazard and the implementation of them in EUNADICS EWS, have been investigated, showing added value to current existing public system (in the late 2010s). Note that the observation of nuclide cloud is more sensitive and inaccessible to classic users as no data are public. After our review and the possibility with partners, a selection of observations has been determined with respect to the monitoring of European air space facing a crisis related to airborne hazard. Tables 1 and 2 show respectively the inventory of the selected products from satellite, in situ and ground-based instrumentation, to be considered by EUNADICS EWS.

**Table 2: Inventory of NRT satellite products.**

| Quantity | Satellite / instrument | Responsible institution / Provider | Spectral range | Selective alert | Data access |
|---|---|---|---|---|---|
| $SO_2$ VCD | AQUA / AIRS | NASA / AIRES/BIRA | IR | Yes | NRT |
| $SO_2$ VCD | AURA / OMI | NASA | UV-vis | Yes | NRT |
| $SO_2$ VCD | Suomi-NPP / OMPS | NASA | UV-vis | Yes | NRT |
| $SO_2$ VCD | Sentinel-5p / TROPOMI | ESA/EU / BIRA/DLR | UV-vis | Yes | NRT |
| $SO_2$ VCD | MetOp-A & B / GOME-2 | EUMETSAT / DLR | UV-vis | Yes | NRT |
| $SO_2$ VCD | MetOp-A & B / IASI | EUMETSAT / ULB | IR | Yes | NRT |
| $SO_2$ plume height | MetOp-A & B / IASI | EUMETSAT / ULB | IR | Yes (trigg. by VCD) | NRT |
| Aerosol index | AURA / OMI | NASA | UV-vis | No | NRT |
| Aerosol index | Suomi-NPP / OMPS | NASA | UV-vis | No | NRT |
| Aerosol index | Sentinel-5p / TROPOMI | ESA/EU / KNMI | UV-vis | No | NRT |
| Aerosol index | MetOp-A & -B / GOME-2 | EUMETSAT / KNMI | UV-vis | No | NRT |
| Ash index | AQUA / AIRS | NASA / ULB/BIRA | IR | Yes | NRT |
| Ash index | MetOp-A & -B / IASI | EUMETSAT/CNES / ULB | IR | Yes | NRT |
| Aerosol class. (ash & dust) | MetOp-A & -B / IASI | EUMETSAT/CNES / ULB | IR | Yes | NRT (daily data) |
| AOD (desert dusts) | MetOp-A & -B / IASI | EUMETSAT/CNES / ULB | IR | Yes | NRT (daily data) |
| Ash mask | MSG / SEVIRI | EUMETSAT / DLR | broadband | Yes | NRT |
| Ash column load | MSG / SEVIRI | EUMETSAT / DLR | broadband | Yes | NRT |
| Ash top height | MSG / SEVIRI | EUMETSAT / DLR | broadband | Yes | NRT |
| AOD | Terra & Aqua / MODIS | NASA | broadband | No | NRT (24h delay) |
| Ash index | Sentinel-3A & B / SLSTR | ESA / FMI | broadband | Yes | NRT |
| Ash top height | Sentinel-3A & B / SLSTR | ESA / FMI | broadband | Yes | NRT |

For the implementation in EUNADICS EWS, we consider that each selected product can be characterised by four categories of information: the basic information (quantity, instrument/platform, responsible institute/provider, units), the time-space resolution (temporal resolution, spatial/vertical resolution), the data availability information (spatial coverage for satellite and in situ instruments, temporal coverage for in situ instruments, Overpass time at the equator for satellite instruments, time delay for delivery, measurements schedule for ground-based instruments, processing level for satellite, data format, data volume, access, link to product overview,



dissemination/link to data), and the implementation information for data integration and decision-making (alert/notification, visualisation system). Tables 2 and 3 present a subset of all the characterisation established for these products.

**Table 3: Inventory of NRT ground-based and in-situ products.**

| Quantity | Ground-Based instrument | Network | Selective alert | Data access |
|---|---|---|---|---|
| Aerosol extinction coeff. | Lidar | ACTRIS / EARLINET | No | partly NRT |
| Aerosol backscatter coeff. | Lidar | ACTRIS /EARLINET | No | partly NRT |
| Volume depol. Ratio EWS index | Lidar | ACTRIS / EARLINET | Yes | partly NRT |
| Att. backscatter coeff. | Auto. lidars & ceilometers | EUMETNET / E-PROFILE | No | NRT |
| Uncalib. Att. backscatter coeff. | Auto. lidars & ceilometers | EUMETNET / E-PROFILE | No | NRT |
| Aerosol layer altitude | Auto. lidars & ceilometers | EUMETNET / E-PROFILE | No | partly NRT |
| Backscatter coefficient | Lidar | IMO | No | partly NRT |
| Aerosol type | Lidar | IMO | No | partly NRT |
| Reflectivity | X-band radar | IMO / INGV | No | NRT |
| Reflectivity | C-band radar | IMO | No | NRT |
| **In Situ instrument** | | | | |
| Seismicity | SIL seismic network (~80 seismometers) | | Yes (through VONA) | NRT |
| Volcanic tremor | Seismic stations | | Yes (through VONA) | NRT |
| **Radiological data** | | | | |
| External dose rates | Radiation monitoring networks | EURDEP | Yes | NRT |
| Nuclide concentrations | Air sampler networks | EURDEP | Yes | NRT |

### 2.3 Requirements for data integration and harmonisation

Information gathered during the EUNADICS stakeholders workshop (Cologne, October 2017) and discussions with the VAACs at the AeroMetSci conference (Toulouse, November 2017), has brought a precise definition of the automated products needed from models that can be relevant for flight planning and management. A review of the inputs on data integration and harmonisation required by atmospheric transport and dispersion model, has
been established to determine the most relevant product information/format/parameters to be implemented in our EWS. The list of information/format/parameters is the following: satellite and ground-based Aerosol Optical Depth — AOD (from different platforms), satellite and ground-based lidar attenuated backscatter, ground-based measurements of particle matter, volcanic ash plume height (from radars, observatories), volcanic ash total column and plume-height from satellite, aircraft measurements of particle matter, lidar aerosol layer altitude, ground-
based measurements of $SO_2$, satellite $SO_2$ column with average kernel and error estimate, aircraft measurements of $SO_2$, satellite $SO_2$ plume height, $SO_2$ profiles by ground-based spectrometers (observatories), nuclide concentrations from air sampler networks, and external dose rates from radiation monitoring networks. This list has been considered to obtain the inventories of observations in Tables 2 and 3.

### 2.4 Review of external reports

We proceeded a review of external reports (Zehner et al., 2010; ESA VAST user requirements, 2013; WMO SCOPE Pilot Project Criteria, 2017; Inter Pilot Magazine, Issue 1, 2018) to determine the key development of the EWS and the most relevant product that would fulfil the requirement of the users. The same three types of requirements as in section 4.2 have been identified and are presented in Tab. 4. Requirements for the data integration (section 2.3) have been considered to define a list of data product candidates (Tab. 3 and 4) from
inventories of satellite, in-situ and ground-based observations (section 2.2).





**Table 4: Overview of user requirements from external sources.**

| 1. Quality of information | | |
|---|---|---|
| | a) | product delivery (timely and reliable manner) |
| | b) | charts of tailored products (e.g. synergy between different kind of observations) |
| | c) | points of contact for data problems, issues and feedback |
| | d) | NRT data delivery (maximum 6 hours after measurements) |
| | e) | use of multi-source satellite data (on different platforms) is highly required |
| | f) | development of quantitative ash concentration retrievals (combined with model data) |
| | g) | communication (product generation and availability) to integrate novel development in user systems |
| 2. System interoperability | a) | documentation (product characteristics, format, source of origin, algorithm, limitations) |
| | b) | trainings |
| | c) | compatibility and homogenisation of dataset format |
| | d) | global coverage from satellite data (not only over Europe) |
| | e) | validation and certification of satellite products by other sources |
| 3. Improvement of the display (requirements from pilots and airlines) | a) | open and unrestricted access |
| | b) | NRT standardised volcanic ash product with a repetition rate of 15 minutes or better |
| | c) | practical dataset (universal format, colour patterns to highlight ash reports and forecasts) |
| | d) | indication of at least three levels of ash or $SO_2$ contamination (e.g. no relevant, low and high) |
| | e) | 3D display of airborne hazard clouds; e.g. with the use of Google Earth kml files and/or similar format compatible with Skyvector charts (see https://skyvector.com), or with Electronic Flight Bag (EFB), an electronic information management device helping flight crews to perform flight management tasks more easily and efficiently |

Considering user requirements from section 2.1 and 2.4, the requirements tackled by EUNADICS EWS concern:

- the improvement of the quality of information by optimising the risk assessment (using reliable and fast

NRT observations, providing with level of confidence and errors bars if available, and implementing

tailored products), by the use of quality labels to increase the reliability of information, by providing

point of contact and communicating about the data product version and availability with respect to data

integration (points 1.a.i., 1.a.ii. and 1.b in Tab. 1, and all points of 1. in Tab. 4)

- a contribution to the system interoperability by bundling the maximum of available information about

selective detection of airborne hazard, by implementing compatible and homogenised information with

a global coverage, and by providing restricted access for key users to avoid misunderstanding of public

users (points 2.c and 2.d in Tab. 1 and all points of 1. in Tab. 4).

- the improvement of the display of information by the visualising observations of hazard dispersion (point

3b in Tab. 1, and points 3.b and 3.e in Tab. 4).

Note that a documentation (with product characteristics, format, source of origin, algorithm, validation and

limitation of the selected data products by our EUNADICS EWS), has been provided by partners and the

EUNADICS consortium (requirements 2.a and 2.e of Tab. 4).

**3 Existing monitoring and warning systems**

**3.1 Example of systems related to volcanic emission**

A primordial source of information about volcanic activity is in the hand of the Volcanic Observatories, such as

the Icelandic Volcano Observatory operates within the Icelandic Meteorological Office (IMO; https://en.vedur.is)

and the Etna Observatory in Catania operates by the Italian National Institute of Geophysics and Volcanology

(INGV-OE; https://www.ct.ingv.it). IMO and INGV are both partners of EUNADICS. These observatories are



equipped with ground-based instruments for providing as much as possible fast and precise information
concerning active volcanoes. Volcanic activity is monitored by observatories using a combination of fixed
instruments (e.g. seismometers, infrasound arrays, continuous GNSS stations, strainmeters, gas detectors, river
monitors, radars) in addition to mobile instruments (e.g. mobile radars and lidars). Such instrumentation can
stream live data from the field to the office, bringing critical information to the control cells. During episodes of
volcanic unrest or eruption, observatories provide information to Civil Protection authorities, local populations
and notifications to the VAACs. This procedure was established by ICAO, for providing crucial information
during volcanic eruptions to the aviation sector, and both IMO and INGV are using it. Volcanic Observatories
provide notification of eruptive activity using the Volcano Observatory Notices for Aviation (VONA) messages
that are issued according to the ICAO Doc.9766-AN/968 "Handbook on the International Airways Volcano Watch
(IAVW)" (ICAO, 2012, 2014a, 2014b; Lechner et al., 2018). The VONA messages are aimed at dispatchers,
pilots, and air-traffic controllers to inform them about volcanic unrest and eruptive activity that could produce
ash-cloud hazards. As an example, for Etna volcano, the VONA messages are sent by the Control Room of INGV-
OE, which operates on a 24/7 basis, and they can be downloaded, together with other bulletins, reports, tremor
graphs, images from video surveillance network, volcanic ash dispersal, etc. For the monitoring of Icelandic
volcanoes, a network of UV spectrometer is used. This means that, for an event for which an increase of $SO_2$ is
deemed to be related to magma movement, an alert is issued by the VONA. The same is applied to the seismicity;
i.e. if there is a significant increase in seismicity (intense seismic swarms) or seismic tremors, this is outlined in
the VONA and relayed through EWS. About regional monitoring, we can notice the Kamchatka Branch of
Geophysical Survey (from the Russian Academy of Sciences) and the Kamchatka Volcano Eruption Response
team (KVERT). A web interface allows to show the activity in this region ([www.emsd.ru](http://www.emsd.ru)). Information about the
plume height are retrieved in NRT using camera, and email notification are sent to the VAACs and researchers.
Concerning a specific support to aviation with a global coverage of possible volcanic emission, as far as we know,
we can mention three existing EWS. The NOAA/CIMSS (US National Oceanic and Atmospheric Administration
/Cooperative Institute for Meteorological Satellite Studies) VOLcanic Cloud Analysis Toolkit (VOLCAT) web
site features NRT processing of many geostationary and low-earth orbit satellites covering much of the globe
([https://volcano.ssec.wisc.edu](https://volcano.ssec.wisc.edu)). VOLCAT includes a collection of sensor agnostic algorithms for detecting,
tracking, and characterising volcanic ash and $SO_2$ (e.g. Pavolonis et al., 2015a, 2015b; Pavolonis et al., 2018;
Hyman and Pavolonis, 2020), and the products are utilised by many of the VAACs. The VOLCAT products are
scheduled to achieve full operational status in NOAA in 2023. The alerting service consists of four categories of
alerts: sudden changes in thermal output (hot spots), newly detected ash emissions, newly detected rapidly
developing clouds near known volcanic vents, and newly detected $SO_2$ emissions. Users can subscribe and
configure alert subscriptions using a web interface ([https://volcano.ssec.wisc.edu/alert](https://volcano.ssec.wisc.edu/alert)). Alerts are shown on an
event dashboard. Access to the alerts and event dashboard is currently restricted to VAACs, MWOs, volcanic
observatories, and research collaborators, as these are considered pre-decisional products. On the other hand, the
SACS EWS is a highly successful system used by agencies worldwide (Brenot et al., 2014). This system, hosted
by one of EUNADICS partners (BIRA; [http://sacs.aeronomie.be](http://sacs.aeronomie.be)), was initiated by the European Space Agency
aims at supporting the Volcanic Ash Advisory Centres, like Toulouse VAAC and London VAAC. NRT data of
$SO_2$ and volcanic ash are derived from hyperspectral sensors onboard polar orbiting satellite, in the ultraviolet-
visible (UV-vis) range with OMI, GOME-2B, GOME-2C, OMPS and TROPOMI, and in the infrared (IR) range





with AIRS, IASI-A and IASI-B. The SACS multi-sensors system addresses automatically worldwide detection of

volcanic plumes of $SO_2$ and ash notifications, sending alert by email to interested parties (https://sacs.aeronomie.be/alert). Finally, a continuous analysis and a systematic surveillance is in operation at the Free University of Brussels (ULB) in order to detect a possible anomalous threshold of $SO_2$ caused by a volcanic eruption. This automatic system, based on IASI data (onboard MetOp-A & -B & -C), sends email alerts to the VAACs and key users when high $SO_2$ levels are detected (http://cpm-ws4.ulb.ac.be/Alerts). Information about

$SO_2$ column density and layer height is provided. This IASI detection system provides automatically inputs of $SO_2$ (and ash) products to SACS system, aiming at providing NRT $SO_2$ and ash measurements related to volcanic emissions.

**3.2 Example of systems related to dust and sandstorms**

The monitoring of extreme dust events is critical for aviation. Amongst existing system, we can mention the WMO

Sand and Dust Storm Project, initiated in 2004, and its Sand and Dust Storm Warning Advisory and Assessment System (SDS-WAS), launched by the Fifteenth World Meteorological Congress in 2007. SDS-WAS enhances the ability of countries to deliver timely, quality sand and dust storm forecasts, observations, information and knowledge to users through an international partnership of research and operational communities. This service is divided into three regional centres. The WMO SDS-WAS Regional Centre for Northern Africa, Middle East and

Europe (https://sds-was.aemet.es), is coordinated by a Regional Centre in Barcelona, Spain, and hosted by the State Meteorological Agency (AEMET) and the Barcelona Supercomputing Centre (BSC). The WMO SDS-WAS Regional Centre for Asia (http://www.asdf-bj.net), is coordinated by a Regional Centre in Beijing, China, and hosted by the China Meteorological Administration (CMA). The WMO SDS-WAS Regional Centre for the Americas (http://sds-was.cimh.edu.bb), established in the USA with a possible regional centre hosted by the

Caribbean Institute for Meteorology and Hydrology (CIMH) in Barbados, focuses on the health implications of airborne dust. The prime objective of the three SDS-WAS regional centres is to lead the development and implementation in the region of a comprehensive system for mineral dust observation and forecast, with special emphasis on extreme sand and dust events. Theses observational systems aim to a continuous dust monitoring, validation and verification of forecast products and data assimilation into numerical models. SDS-WAS models

used include ground observations (particulate matter measurements progressively becoming available in NRT, indirect information from regular weather reports and remote-sensing retrievals from sun photometers or vertical profilers) and satellite products (single-band images, qualitative multi-band products designed to improve dust identification or quantitative retrievals). Currently, the WMO SDS-WAS Regional Centre for Northern Africa, Middle East and Europe provides a multi-model platform with analysis and +54 hours forecasts for 12 dispersion

models (Nickovic et al., 2001; Woodward et al., 2001; Zakey et al., 2006; Benedetti, et al., 2009; Morcrette et al., 200; Colarco et al., 2010; Pérez et al., 2011; Haustein et al., 2012; Basart et al., 2012, 2020; Lu et al., 2016). SDS-WAS contributes to the International Cooperative for Aerosol Prediction (ICAP), an unfunded international forum for aerosol forecast centres, remote sensing data providers, and lead systems developers, which coordinates the first global multi-model Ensemble for aerosol forecasts, as described in Sessions et al. (2015). The use of the


multi-model system is overall better than any individual model. Over specific regions, combining several models
       leads to better forecasts than the best individual model even when number of ensemble members is small.

### 3.3 Example of systems related to smoke from wildfires and biomass burning

NASA provides a global EWS related to fire detection from its Fire Information for Resource Management System
(FIRMS). It provides Fire Radiative Power (FRP) from Low Earth Orbit (LEO) satellites sensors, i.e. MODIS
instruments onboard the Terra and Aqua satellites (Kaufman et al., 1998; Giglio et al., 2003; Justice et al., 2011)
       and VIIRS sensor onboard Suomi-NPP (Csiszar et al., 2014). FIRMS focus and objectives include providing
       quality resources for fire data on demand, working with end users to enhance critical applications assisting global
       organisations in fire analysis efforts, delivering effective data presentation and management
       (https://firms.modaps.eosdis.nasa.gov/alerts). On the other hand, CAMS has developed a monitoring system,
which provides observations of fire detection and forecasts of smoke dispersion. By using NRT observations of
       the location and intensity of active wildfires, i.e. FRP product based on SEVIRI (Roberts et al., 2015) from the
       EUMETSAT LSA-SAF (http://landsaf.meteo.pt), CAMS estimate the emissions of aerosols and pollutants. This
       is done through its Global Fire Assimilation System (GFAS). This allows active fires to be monitored and their
       estimated emissions to be used in the CAMS forecasts to predict the transport of the resulting smoke in the
atmosphere (https://atmosphere.copernicus.eu/fire-monitoring). The forecasts are used in air quality apps, to help
       people limit their exposure to pollution, and by policymakers and local authorities to manage the impact of fires.
       The Copernicus Emergency Management Service (EMS) has developed the European Forest Fire Information
       System (EFFIS; http://effis.jrc.ec.europa.eu; see EFFIS, 2018). This system supports the services in charge of the
       protection of forests against fires in the EU and neighbour countries and provides the European Commission
services and the European Parliament with updated and reliable information on wildland fires in Europe.

### 3.4 Example of systems related to radionuclide clouds

This fourth type of system addresses the monitoring of nuclear accidents and radionuclide plumes. The
development of such system is quite sensitive and the dissemination of information is confidential. Most European
dose-rate results are recorded at the European Radiological Data Exchange Platform (EURDEP) web site
(https://eurdep.jrc.ec.europa.eu) but accessing the site and downloading data requires agreements. A collaboration
       with the Joint Research Centre of the European Commission (JRC) is required to establish NRT or archive access
       to data (including historical data). Individual countries can provide their own data (i.e. providing gamma dose
       rates, including spectrometric, and activity concentrations in air). For selected case studies and research, airborne
       activity concentration measurements of radionuclide-bound aerosols may be provided by selected laboratories.
EMERCON (Emergency Convention) messages are also produced by the International Atomic Energy Agency
       (IAEA) through the WMO RSMCs. An EMERCON message is a descriptor referring to the official system for
       issuing and receiving notifications, information exchange and assistance provision through the IAEA's
       Emergency Response Centre in the event of a nuclear or radiological emergency. The ICAO system allows the
       issuance of SIGMET (SIGnificant METeorological Information message) for radioactivity , from the ground to
unlimited flight level (FL), up to 30 km radius from the release site. Since 2020, the Russian Federal
       Environmental Emergency Response Centre (FEERC) has an operational system that provides public
       concentration charts for use in aviation (http://aviamettelecom.ru/activity/methodical). Such charts are issued in





case of hazardous radioactive release according the "Guidance material on the dissemination of information on accidental release of radioactive material into the atmosphere" (FEERC, 2019). An example of chart (from a

radioactivity advisory) showing airspace aviation FL contaminations based on conditional data (e.g. from a radionuclide cloud at Fukushima Daiichi, on 28 May 2019), can be find in the additional material.

**4 Pre-alerting mechanism of EUNADICS EWS**

**4.1 Detection of airborne hazard using satellite**

**4.1.1 Selective detection of volcanic SO₂**

The impact of the $SO_2$ exposure and the sulphur damage to engines has affected hundreds of flights in the last decade. Sulphidation mechanisms can cause damage to the engine with solid diffusion process or corrosion-fatigue. A flight through a volcanic plume and the exposure to $SO_2$ is a problem for passengers and aviation stakeholders as it is a threat to the safety and health, and it requires turbine maintenance. The detection of $SO_2$ from satellite is straightforward (see Fig. 1), and of great interest for aviation. Generally, wAhen $SO_2$ clouds reach

the free troposphere up to the lower stratosphere, it is a good indicator of volcanic activity. Height satellite sensors are considered by EUNADICS EWS for the $SO_2$ detection, that is retrieved by OMI, GOME2-B, OMPS and TROPOMI in the UV-vis (Yang et al., 2007; Rix et al., 2009, 2012; Li et al., 2017; Theys, 2017, 2019), and AIRS, IASI-A and IASI-B in the IR range (Prata and Bernardo, 2007; Clarisse et al., 2008, 2012; Clerbaux et al., 2009). Details about the detection and the limitation of these products can be find in Brenot et al. (2014).

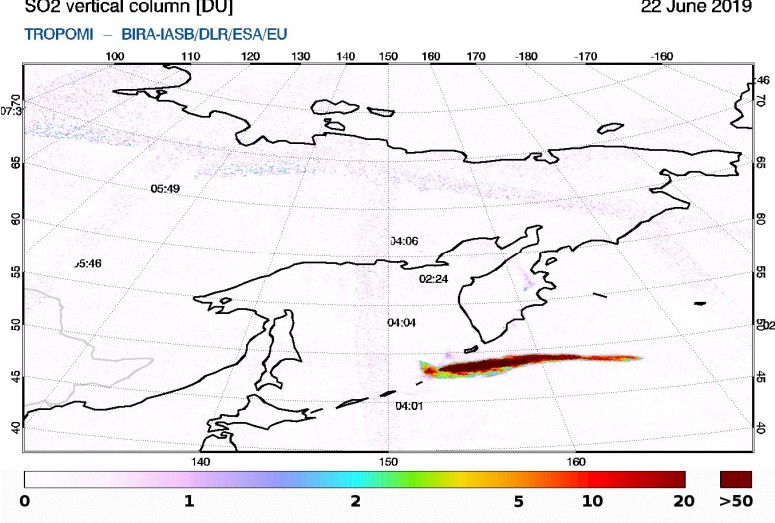

**Figure 2: SO₂ detection from TROPOMI on 22 June 2019 (Raikoke eruption, Kuril Islands).**

Satellite IR sensors considered by EUNADICS (IASI-A & -B) can measure the $SO_2$ layer height (SO2LH) using optimal estimation algorithms (Clarisse et al., 2014). The SO2LH retrieval is very fast, with an accuracy of 1-2 km, which can be obtained even for low $SO_2$ column (under 1 DU). SO2LH results are obtained for estimates

between 3 and 21 km, with low performance for heavily saturated plumes. Figure 3 shows SO2LH from IASI-A on 23 June after Raikoke eruption.

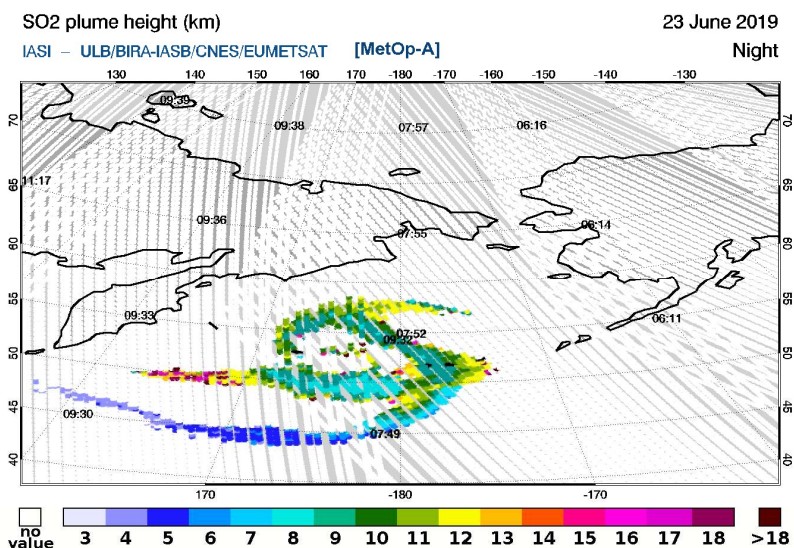

**Figure 3: Characterisation of Raikoke plume by SO$_2$ layer height from IASI-A on 23 June 2019.**

**4.1.2 Selective detection of ash**

The volcanic ash detection from satellite is far from trivial but is essential for aviation, as it can cause severe damage to turbine engines (Clarkson et al., 2016). Differential absorption by ash between two channels, e.g. by using of the brightness temperature difference (BTD) between the 11 and 12 μm channels for SLSTR instrument, can be used to detect volcanic plume, as illustrated in Fig. 4. BTD results from SLSTR are expressed in K. A negative value of -2K issue an alert for a data granule (3 granules with data for a time duration of about 3 minutes

are shown in blue in Fig. 4).

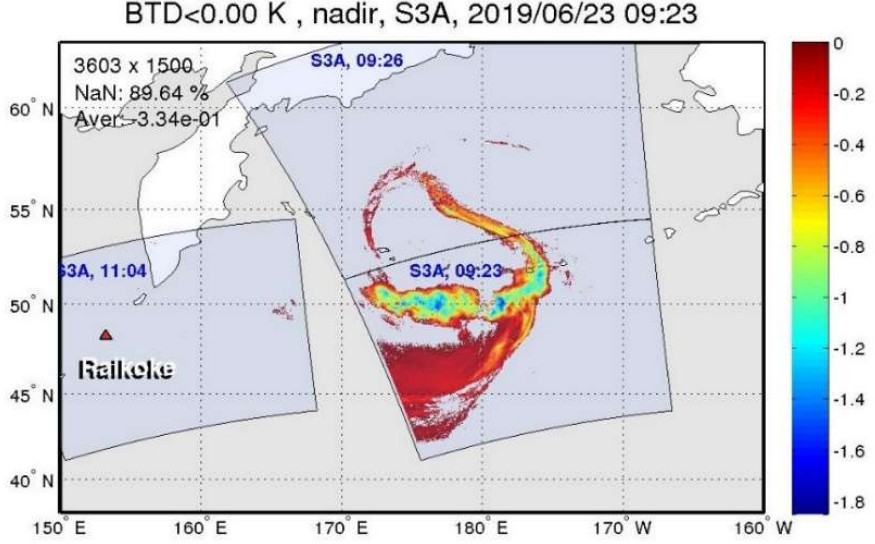

**Figure 4: Ash index (diff. Brightness Temperature, in K) from SLSTR sensor onboard Sentinel-3A on 23 June 2019.**

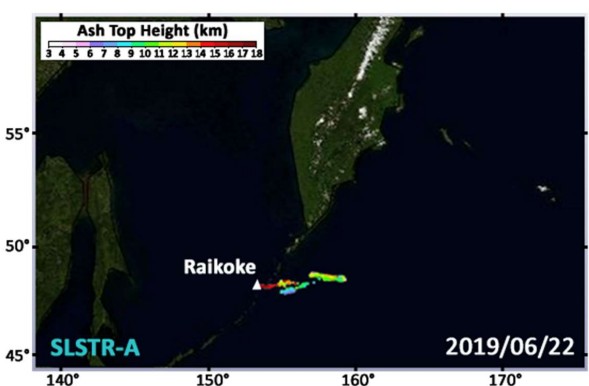

**Figure 5: Characterisation of Raikoke plume by Ash Top Height (in km) from SLSTR-A on 22 June 2019.**

The BTD technique can yield to false detection in the presence of absorbing aerosols other than ash (e.g., desert dust). However, using SLSTR sensors (onboard Sentinel-3A & -3B), combining automated aerosol detection, the algorithm of Virtanen et al. (2014) can provide ash/dust height estimate (Fig. 5). This algorithm is based on matching top of the atmosphere reflectances and brightness temperatures of the nadir, and 55 forward views, and using the resulting parallax to obtain the height estimate (stereoscopic technique).

The absorption signatures of ash can be discriminated from other types of absorbing aerosols using hyperspectral thermal infrared sensors (Clarisse et al., 2010). The IASI-A&B and AIRS ash index products used in EUNADICS EWS are based on a three-step process (Clarisse et al., 2013), which provides ash detection with levels of confidence. Four colour-codes are used to defined the level of confidence of pixel ash detection. High level detection appears in red (ash is almost certainly present with less than 1% of false alerts; this issues an alert), and

Medium level in orange (ash detection with high confidence nevertheless no alert is issued). Low level is in yellow (mineral aerosol signature observed in spectra and proximity to a high confidence detection), meaning the volcanic ash detection is highly probable but false detections can still occasionally occur. The unknown detection appears in white (no ash detected by the algorithm). Figure 6 illustrated a selective detection of ash from AIRS few hours after the start of Raikoke eruption.

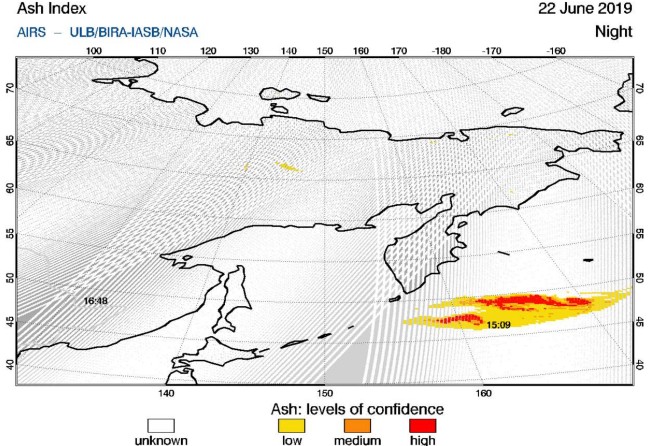


**Figure 6: Ash detection from AIRS on 22 June 2019 (Raikoke eruption, Kuril Islands).**





For the dedicated EU area, a last ash detection from geostationary satellite is considered by EUNADICS EWS, i.e. the ash mask, column load (in kg/m²) and top height (in km) products from SEVIRI (onboard MSG). The VADUGS (Volcanic Ash Detection Utilising Geostationary Satellites) algorithm used to retrieve these products,

is based on a backpropagation neural network which combined BTD technique with mask of high clouds and atmospheric model look-up tables for a broad range of particle concentrations for different ash types in various layers (Kox, 2012; Kox et al., 2013; Graf et al., 2015). Figure 7 presents ash column load on 13 May 2010 (Eyjafjallajökull eruption, Iceland). An ad-hoc version of this algorithm tailored for Himawari has also been developed and applied to the Sinabung 2018 eruption (de Laat et al 2020).


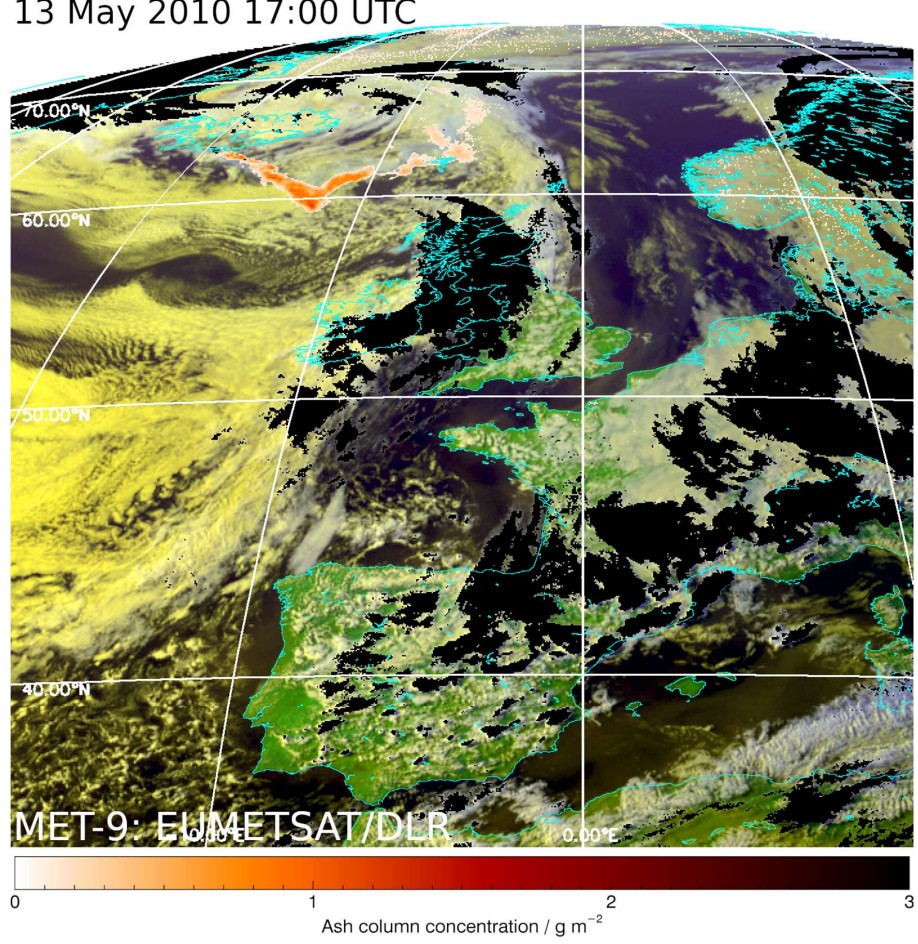

**Figure 7: Ash column concentration from SEVIRI on 13 May 2010 (Eyjafjallajökull eruption, Iceland). Orange-red: ash-detection (column concentration), black: cirrus mask.**



**4.1.3 Selective detection of dust**

Clarisse et al. (2010, 2013) have demonstrated the potential of hyperspectral thermal infrared sounders to detect volcanic ash with a high sensitivity and differentiate it from other airborne aerosol (including windblown sand), as illustrated in Fig. 8.

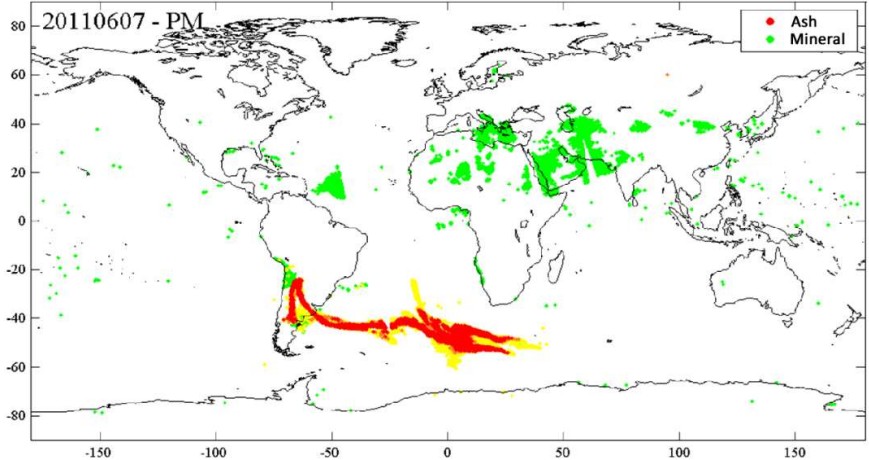

**Figure 8: IASI-A detection of volcanic ash (Puyehue-Cordón Caulle eruption) and mineral dusts (windblown desert sands) on 7 June 2011 (afternoon).**

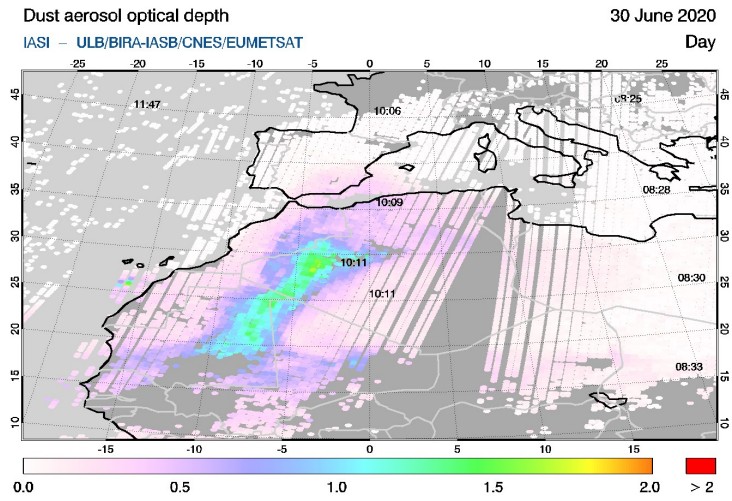

**Figure 9: Saharan dust AOD observed by IASI-C on 30 June 2020.**

Clarisse et al. (2020) have recently presents a new algorithm for estimating atmospheric dust optical depths (dust

AOD) and associated retrieval uncertainties. This AOD retrieval, specific to dust, are based on the calculation of a dust index and on a neural network trained with synthetic IASI spectra. It has an inherent high sensitivity to dust and efficiently discriminates mineral dust from other aerosols, as show in Fig. 8 (see Clarisse et al., 2013, 2020). EUNADICS EWS considers dust flag and dust AOD from IASI (onboard MetOp-A, MetOP-B and MetOp-C). As an example, Saharan dust AOD is observed by IASI-B on 30 June 2020 (Fig. 9).




### 4.1.4 Triggered detection of dust or smoke


This approach essentially relies on the collocation (or near collocation) of a selective product with a non-selective product to trigger (and propagate) alerts. Figure 10 illustrates the triggered approach applied by EUNADICS EWS to obtain detection of dust from the combined Dark Target and Deep Blue AOD products from MODIS-Terra (Levy, R., Hsu, C., et al., 2016) using selective detection of dust from IASI-A (colour code red for dust flag = 1).

This figure shows all the AOD pixel values from MODIS (for a data granule measuring between 11:20 and 11:25 UTC on 15 October 2017). A threshold value is used to select the AOD from MODIS (threshold of 0.7) for testing geographic matching (critical distance of 200 km) with alert pixels (from IASI-A). These alert data pixels have been obtained from a previous AOD alert from IASI-A, and are stored by EUNADICS EWS as active dust alert pixels (defining the pool of dust alerts). Each pixel of this pool is characterised by a position and a time. An alert

pixel of the dust pool stays active during 12 hours. If a match is obtained between selected MODIS AOD pixels and the alert pixels from the dust pool, the MODIS AOD pixel is determined as an alert pixels (which join the dust pool). A completion of the dust cloud from MODIS AOD is operated looking at the neighbouring pixels (with AOD value over 0.5) of all the newly obtained dust alerts (completed cloud in Fig. 10).

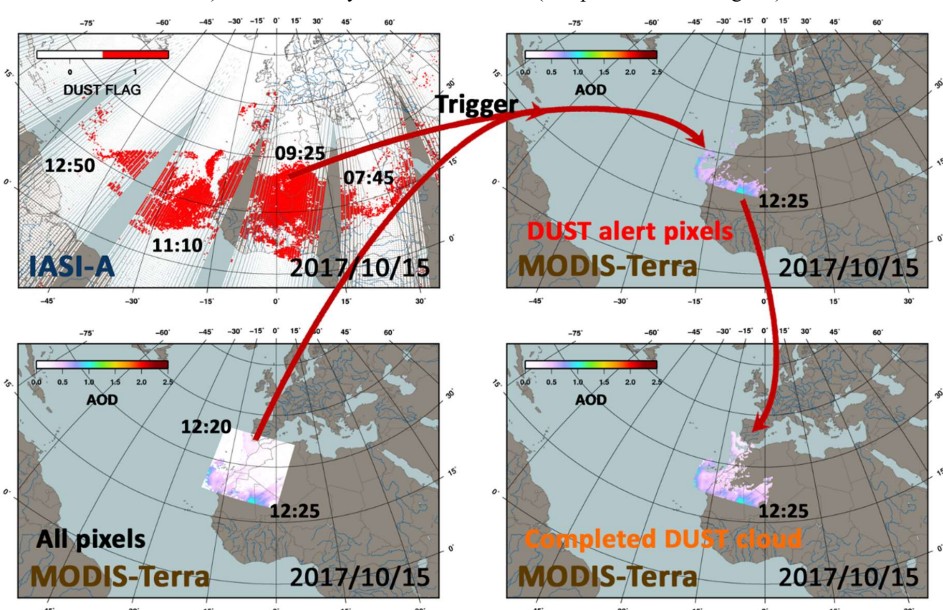

**Figure 10: Illustration of the triggered approach to isolate MODIS-Terra AOD pixels related to desert dust (2017/10/15) using dust flag from IASI-A. Completed AOD data related to the dust cloud can be retrieved.**

On 15 October 2017, the Hurricane Ophelia reached Ireland (downgraded to storm) and during its northward movement brought desert dust particles and smoke particles from raging fires over the Iberian Peninsula in Europe (Osborne et al., 2019). During this twin event, OMPS instrument measured high values of AAI (between 12:50

and 14:40 UTC) due to dust from Africa's Saharan desert and smoke from wildfires in Spain and Portugal. Figure 11 illustrates the triggered approach applied to obtain selective detection of dust and smoke cloud from OMPS AAI using, respectively, the pools of dust alert (fed by IASI-A and MODIS-Terra alerts) and smoke alert. The active pool related to smoke is fed by the NRT fire localisation from NASA-FIRMS, i.e thermal anomaly and FRP data from VIIRS sensor, in the present case.

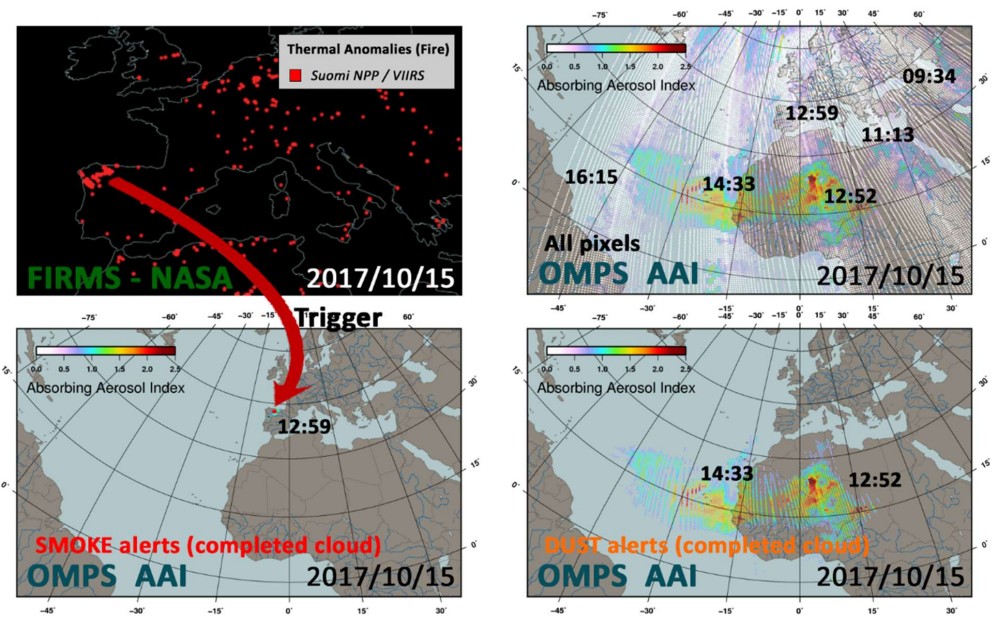


**Figure 11: Illustration of the triggered approach to isolate OMPS AAI pixels related to smoke and dust (2017/10/15).**

EUNADICS EWS identifies smoke alerts from OMPS on 15 October 2017 (Fig. 11). These alert pixels remain active for 24 hours in our system. On 16 October, the smoke pool trigger new OMPS smoke alerts, which are observed over France and UK (Fig. 12). In the same time, the system isolates dust alerts from OMPS, which are

located over the Saharan desert and the Atlantic Ocean. Using both alert pools (dust and smoke), the displacement of the two hazardous clouds (on the 16th and the 17th of October 2017), is shown in Fig. 12. On 17 October, the smoke plume and the respective OMPS alert pixels are retrieved over northern Europe. In the same time, dust alert pixels are still observed over the Saharan desert and the Atlantic Ocean.

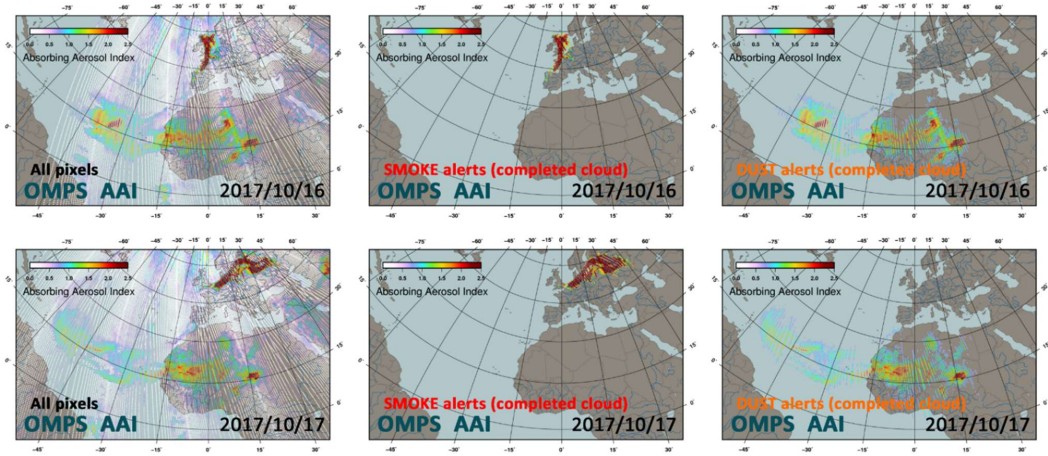

510          **Figure 12: Isolated OMPS AAI pixels related to dust and smoke on 2017/10/16 and 2017/10/17.**



### 4.2 Detection of airborne hazard using ground-based network

### 4.2.1 Selective detection of aerosols hazard using EARLINET network

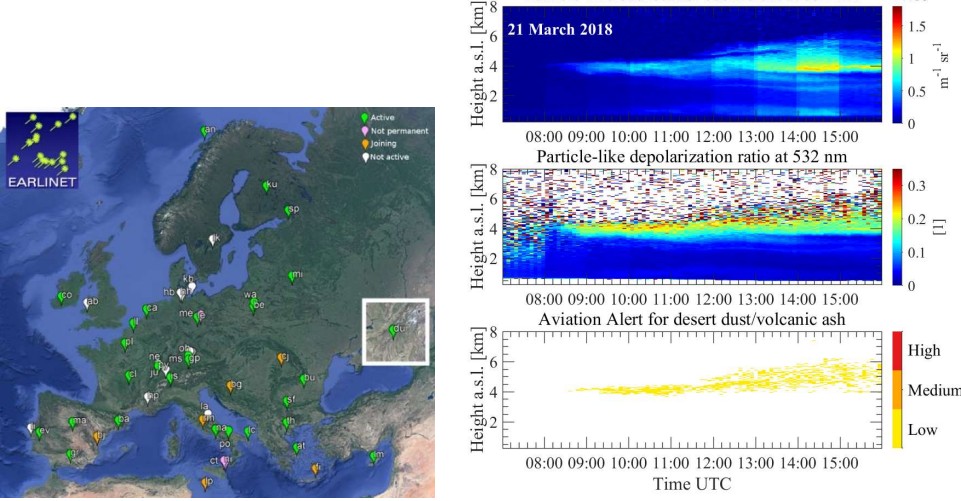

**Figure 13: (left) Map of EARLINET network (from https://www.earlinet.org), and (right) illustration of desert dust / volcanic ash alerts (station Finokalia) during desert dust storm on 21 March 2018 (particle backscatter coefficient at 532 nm, particle depolarization ratio at 532 nm, and alert for aviation.**

A tailored alerting product from the European Aerosol Research Lidar Network (EARLINET; https://www.earlinet.org) has been developed during the EUNADICS project (Papagiannopoulos et al., 2020). It has been designed for NRT EWS applications. Using a stand-alone lidar-based method for detecting airborne hazards (volcanic ash and desert dust; no discrimination), this product is based on the EARLINET Single Calculus Chain (version 5.1), and provides temporally high-resolved, calibrated attenuated backscatter and volume depolarisation ratio (at 532 nm), and cloud mask. The vertical resolution is 7.5 m, and the temporal resolution is 30 s. From these calibrated data, further particle-like products (particle-like backscatter and depolarisation ratio) can be retrieved that act as the basis of the tailored product. The final product (to be used by EUNADICS EWS) is the aviation alert for desert dust/volcanic ash with a three colour-codes. This alert product uses particle mass concentrations (pmc) based on backscatter coefficient thresholds. High level detection appears in red (almost certain detection of ash or dust aerosol with pmc $\geq$ 4 mg/m³), Medium level in orange (4 mg/m³ > pmc $\geq$ 2 mg/m³). Low level is in yellow (2 mg/m³ > pmc $\geq$ 0.2 mg/m³). An example of alert from EARLINET is shown in Fig. 13. On 21-22 March 2018, the eastern Mediterranean, in particular Crete island, was under extreme effects of warm southerly winds pushing enormous amounts of Saharan dust – and hot air – northwards. The desert dust storm caused the closure of Heraklion airport (Crete, Greece) on 22 March 2018, and was, in particular, detected by the ground-based LIDAR from EARLINET.



Natural Hazards
and Earth System
### 4.2.2 Quicklook images from E-PROFILE network

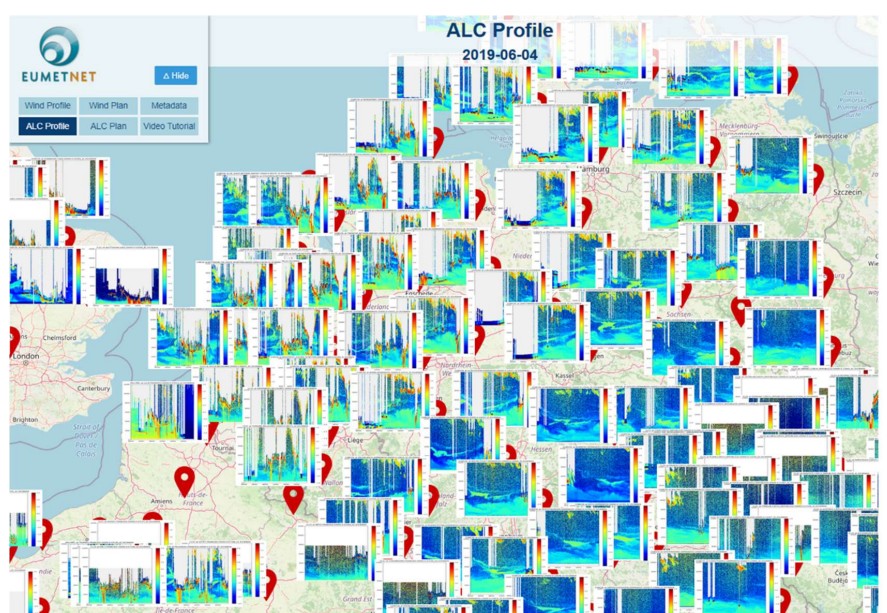

**Figure 14: The E-PROFILE ALC network with quicklooks of attenuated backscatter observations from https://e-profile.eu. The quicklooks have time (0:00-24:00 UTC) on their x- and altitude above sea level (0-12 km) on their y-axis. The high values of attenuated backscatter correspond to clouds (orange and red) and aerosols (light blue, green and yellow). Low and mid-level clouds usually completely attenuate the signal above so that no observations above can be reported what is seen by the white areas in the quicklooks.**

In addition to the NRT distribution of its observation data in NetCDF format, the E-PROFILE network from EUMETNET provides an interactive geographical overview of observations. Quicklooks and interactive plots of aerosols and clouds from the network of Automatic Lidars and Ceilometers (ALC, currently 345 units in continuous operation) as well as of wind profile observations from radar wind profilers (40 units in continuous operation) and from precipitation radars (96 units) can freely be accessed at https://e-profile.eu (Haefele et al., 2016). An example is given in Fig. 14 showing the time series of attenuated backscatter profiles for 4 June 2019 over a region of the United Kingdom, France, Belgium, the Netherlands, Germany and the Czech Republic. On this day smoke originating from North American wildfires was present in the region. Besides the aerosols of the planetary boundary layer, the smoke plumes in the free troposphere between 2000 and 6000 m are clearly visible. Here, an example of smoke is shown because such large-scale ash advections have not been seen observed since the establishment of the network. However, all aerosols, including ash (some smaller cases of Etna exist, not shown), can be observed by these instruments. Thanks to the continuous monitoring of ALC and the geographical density of stations in E-PROFILE such events are easily captured when the plume passes over the observation sites. The event of 4 June 2019 nicely demonstrates the value of this network for monitoring the exact altitude and timing of plumes of ash and other hazardous aerosols over Europe in NRT.

### 4.2.3 Detection from regional network

Essential information about volcanic plumes is provided by region networks. EUNADICS EWS is fed by NRT tailored products from two volcanic observatories: IMO in Reykjavik (https://en.vedur.is) and INGV-EO in


Catania (https://www.ct.ingv.it). Precursors of volcanic eruptions and selective detections characterising on-going
events are provided to the EUNADICS system. Relevant information from ground-based and in-situ
instrumentations (e.g. seismic networks, UV- & IR- and web cameras, radars, UV spectrometers, lidars, gas
sampling) and satellite products (e.g. UV&IR sensors, broadband imagers) are used by IMO and INGV to create
and send messages, the so-called VONA (Volcano Observatory Notices for Aviation). The EUNADICS system
ingests the information from VONA to characterise the source of the natural airborne hazard (volcano). Figure 15
shows an example of a VONA message created by INGV-EO. The volcanic observatories use different types of
cameras (video surveillance, thermal or UV cameras) and provide information about the source of the on-going
volcanic eruption (e.g. height and opacity of the plume, quantity of gas emitted, height). This takes into account
information from INGV seismic network (nearby Etna and Stromboli volcanoes). This infrastructure of mobile
and permanent seismometers has an important number of permanent seismic stations operating in the Etna area
(more than 40 stations, with 18 stations located at medium-high volcano) to ensure adequate coverage even of
low magnitude seismicity, and for the detection of seismic-volcanic events. This also considers the automated
scanning FLAME (FLux Automatic MEasurements) network of nine UV scanning spectrometers, which are
installed on the flanks of Mt. Etna, delivering continuously $SO_2$ concentrations and fluxes (Salerno et al., 2009).
IMO provides to EUNADICS EWS a selection of links (images from web cameras) for each monitored sites
(about 30 in total). Cameras are showing in NRT, with few minutes of delay, snapshots (with scale of height) for
several active volcanoes. IMO also provides links to quicklooks from satellites. The focus area is Iceland and
surroundings. Animations from SEVIRI (on board Meteosat-11) geostationary sensor and AVHRR (on board
NOAA-19 and MetOp-A), VIIRS (on board Suomi-NPP) and MODIS (on board Terra) polar orbiting sensors are
available in NRT.

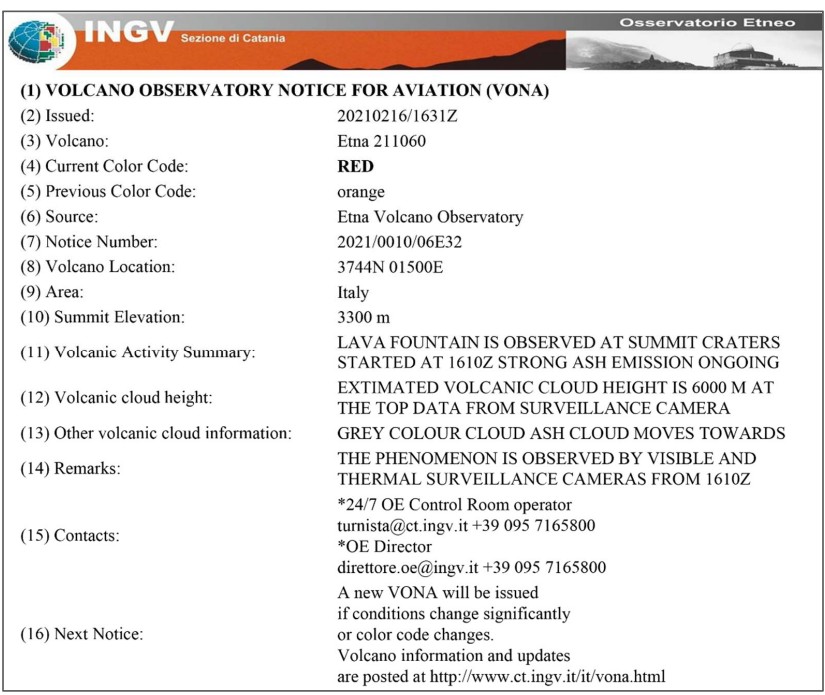

**Figure 15: VONA message for Etna activity (RED) on 16 February 2021 (at the start of a paroxysm episode).**
The IMO volcanic observatory has recently improved the VESPA (Volcanic Eruptive Source Parameter Assessment) system. In case of an on-going eruption, information about the plume height and mass eruption rate from weather radars (C-band, X-band mobile; http://brunnur.vedur.is/radar/vespa), is provided to EUNADICS EWS, which ingests such information (links to quicklooks of the top height of volcanic plume; i.e., 2D images and time-series). An example of the ash plume detected by a mobile X-band radar, during the Grimsvötn 2011 eruption, is displayed in Fig. 16. Data file describing the plume height are also available in NRT, providing the height of the highest point where a significant radar reflection is detected within 10 km distance of the volcano, and the height of the next radar elevation angle above volcano, where plume was not detected. Arason et al. (2011) and Petersen et al. (2012) present more details about the radar and webcam products.

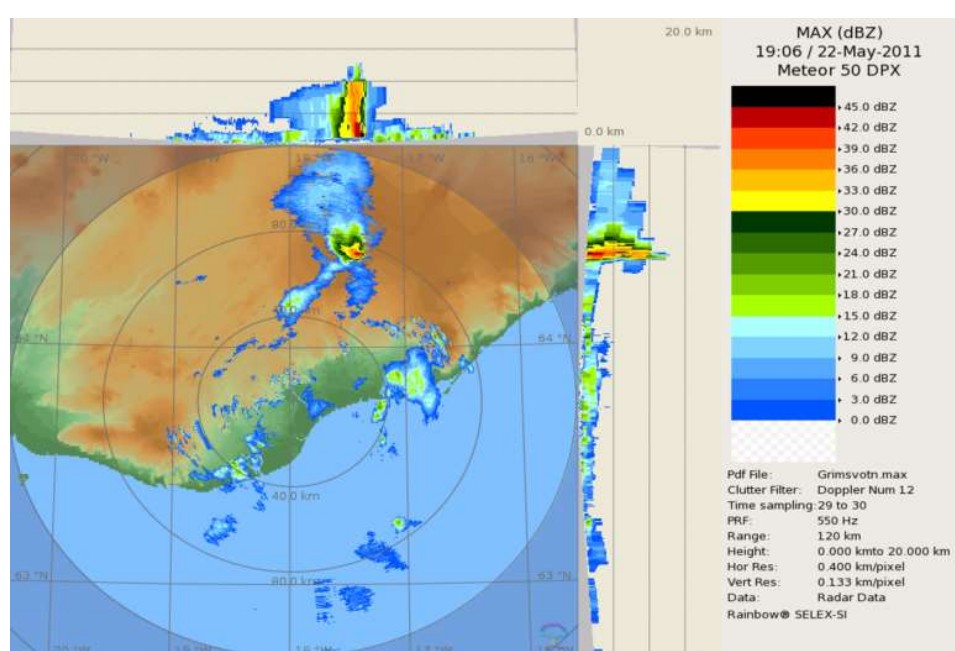

**Figure 16: Radar reflectivity plot displaying ash dispersal during the Grimsvötn 2011 eruption in Iceland.**

### 4.3 Nuclear hazard and EU network

Several sources of radioactivity monitoring data are available, and to obtain robust, harmonised and real-time data is a challenge for the nuclear events. Although not validated Gamma Dose Rate (GDR) radiological monitoring data from most European countries is available in NRT from the EURDEP system (European Radiological Data Exchange Platform, https://eurdep.jrc.ec.europa.eu), the usage of such data inside an automatic notification system common in all the European domain is not done. However, EUNADICS, with view of a future operationalisation, requires access to real-time well established data and data channels. Providing NRT availability and accessibility to such data is complex. The monitoring information is collected from automatic surveillance systems in 39 European countries. Without a radiological event, these data provide information on the background radioactivity and its variability. However if a nuclear accident with gamma emitting radionuclides occurs (e.g. anomalous GRD values over 0.5 µSv/h recorded), the network must be able to capture the existence and the geographical distribution/evolution of the event within the limitations of the network.


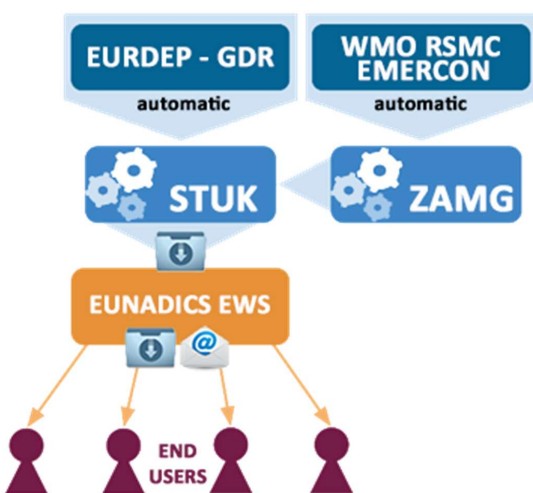


**Figure 17: Approach taken for the interface between radiological data and the final notification to end users.**

The approach taken to design and implement a notification system for nuclear events has largely relied on already existing capacities of the two main contributors to the current work, i.e. ZAMG and STUK. By making use of their national and international mandates and capacities, STUK and ZAM design the interfaces from monitoring

data to notification and posterior alerting have been designed. As illustrated in Fig. 17, the current approach is based on both EURDEP data and the EMERCON (Emergency Convention) messages produced by IAEA through the WMO Regional Specialised Meteorological Centres (RSMCs). Both data sources are ultimately released by STUK after proceeding the filtering of the EURDEP data implemented and collecting the EMERCON information (from ZAMG). STUK finally provides the information to the automatic alerting system developed in EUNADICS.


```
Notification:
Doserate value 2964.960 µSv/h found 2018-09-05 08:00:00 at ATWienSteinhof
Wien-Steinhof at longitude/latitude point (16.2719,48.2064) while
background is 0.100 µSv/h.
Doserate value 86.868 µSv/h found 2018-09-05 08:00:00 at ATRudolfshgel
Rudolfshägel at longitude/latitude point (16.3556,48.1669) while
background is 0.100 µSv/h.

Detailed information:            https://docs.google.com/uc?id=444

This is an automatically generated alert message. Please confirm elevated
dose rate measurements from EURDEP Web Site.
```

**Figure 18: Example of the EURDEP derived alert message.**

Following its national responsibilities, STUK has implemented an elevated GRD alerting software based on measurements available in the EURDEP system. It consists of separate Python scripts that handle data collection,

alerting, and database maintenance functions. This work at national level has been further extended to fit the purpose of EUNADICS EWS. The alerting condition is checked in 10 minutes intervals. A nuclear threat is declared and an alert is created if two nearby measurements show dose rate values higher than the calculated background. Alerting based on a single measurement is not advisable because very often there are high dose rate values caused by other causes, such as broken Geiger Müller tubes, for instance. When an alert in GRD data is





identified, a text message file is created. This alert message is transferred to ZAMG-hosted ftp server, and an email notification is sent to hosting platform of EUNADICS EWS at BIRA. This message transfers data information and a link to a HTML page, which points an interactive map automatically zoomed to the alerting stations. An example of the EURDEP notification, taken from the EUNADICS March 2019 exercise (c.f. the example of the Etna eruption exercise of Hirtl et al., 2020a), is shown in Fig. 18.


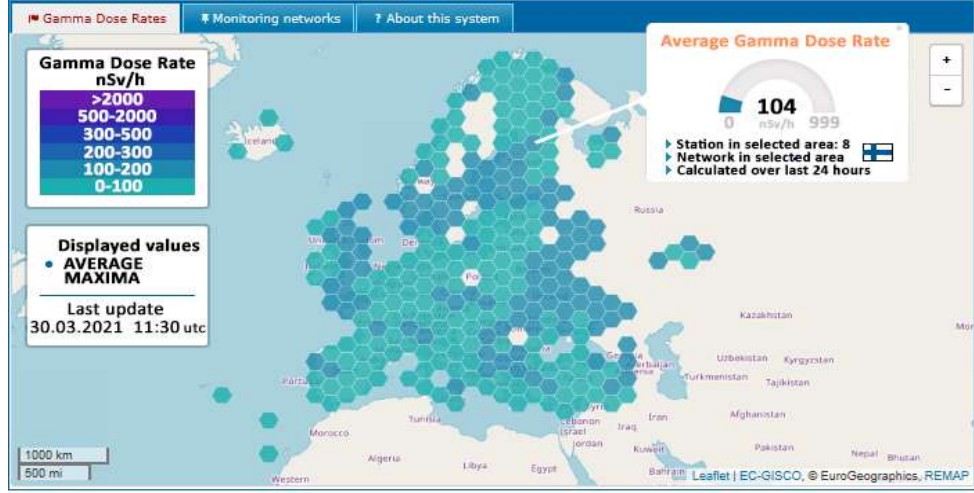

**Figure 19: Link to EURDEP network measurements (Gamma Dose Rates) from EMERCON message.**

The detailed information link leads to a HTML webpage containing the map with zoom and pan features (Fig. 19). The map contains current real online measurements of the EURDEP network. However, and as a note of

caution, currently in the EURDEP system, most European countries do not share validated radiological monitoring data in NRT and gaps may appear per country or time window. In case of GDR alert, an alert data product is created by EUNADICS-AV EWS, in a homogenised format (see next section 4.4.3).

**4.4 Overview of EUNADICS EWS**

The EUNADICS EWS is built on the SACS system (Brenot et al., 2014), which provides NRT satellite data

products (SO$_2$ columns, ash and aerosol index) to aviation stakeholders in the forms of maps and email notifications. The SACS system and its web interface (https://sacs.aeronomie.be) is dedicated to volcanic hazard. The development achieved by EUNADICS concerned an upgrade of SACS system (volcanic emission), and the extension to other airborne hazard (dust, smoke and radionuclide clouds), with creation of news alert products. EUNADICS EWS is a prototype multi-hazard system which has expanded SACS system and create new

functionalities (based on existing mechanisms of SACS) by using: (1) key satellite products from IASI/MetOp-A&-B&-C, SEVIRI/MSG, TROPOMI/Sentinel-5 Precursor, SLSTR/Sentinel-3A&B and MODIS-Terra & -Aqua sensors, (2) ground measurements from EU networks (EARLINET and E-PROFILE) and regional networks from volcanic observatories (IMO and INGV-EO), (3) existing services (VONA messages, EURDEP & RSMC EMERCON messages, NASA-FIRMS). A new functionality obtained by EUNADICS prototype EWS is (4) the

creation of NetCDF Alert data Products (NCAP file) dedicated to the integration in dispersion model and the interest for ATM stakeholders.


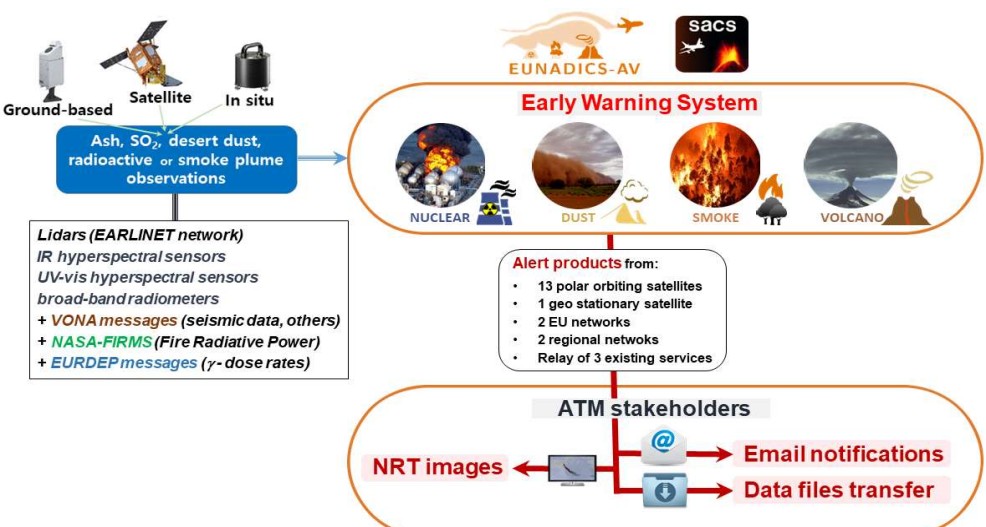

**Figure 20: Illustration of the mechanism of EUNADICS EWS.**

The development of pre-alerting mechanism is based on email notifications (volcanic and radionuclide cloud hazards only) and the creation of homogenised tailored data alert products and alert maps, as illustrated in Fig. 20. Initially, SACS system was built integrating NRT data products ($SO_2$ columns and aerosols/ash indexes) from 7 polar-orbiting instruments (OMI, OMPS, GOME-2A, GOME-2B, AIRS, IASI-A and IASI-B) in a single monitoring and alerting system for volcanic eruptions. This system considers only satellite instruments on board polar orbiting satellites. Figure 20 shows the new additional data information ingested by EUNADICS EWS.

NRT products from satellite, ground-based and in-situ platforms/instruments are provided by EUNADICS partners and external data sources (Tab. 2 and 3), and transferred to the EWS. The blue block in Fig. 20 indicates the multi-source input data (Ash, $SO_2$, dust, radioactive and smoke cloud observations). The structure and the choice of the data products considered in the EUNADICS EWS relies on the user requirements and risk assessments presented in section 2. The automated EWS (including routine data products) applies its own mechanisms to create NRT images and to issue alerts. This represents an extension of the SACS system (Brenot et al., 2014) to other hazards and/or instruments, taking into account inputs from existing systems. Innovative Alert products are created using data from 13 sensors onboard polar orbiting satellites, 1 geostationary imager (SEVIRI onboard MSG9), EU networks (EARLINET, E-PROFILE). 2 regional networks from volcanic observatories (IMO and INGV-EO) and making the relays and uses of 3 existing services (EURDEP, VONA, and NASA-FIRMS). The development of EUNADICS EWS uses inputs from existing EWS addressing continuous monitoring of volcanic ash and $SO_2$ plumes (messages from VONA) and the detection of fires (NASA-FIRMS) and radioactive plume (EURDEP) from NRT observational data portals. In case of detection of exceptional concentrations (possibly threating aviation), notifications are received by EUNADICS EWS, making the relay of these existing system.





### 675 4.5 Mechanism EWS

**Table 5: Overview of EUNADICS alert products from ground-based [GB], satellite [SAT] and in situ [IS] instruments.**

| Quantity | Instrument [GB] [SAT] [IS] | Responsible institute | Source alert | Alert criteria | Access |
|---|---|---|---|---|---|
| Volume depol. ratio | Lidar | ACTRIS/ EARLINET/ CNR | Volcano / Dust | Threshold on volume and particle-like | Off-line |
| Att. Backscatt. coeff. | Auto. lidars & ceilo. | EUMETNET E-Profile | Volc./ Dust / Smoke | Range of att. backscatter | NRT |
| Backscatterig power | Radar | INGV / OPGC | Volcano | Thres. background power | Off-line |
| Backscatterig power | Radar | IMO | Volcano | Thres. background power | NRT |
| Plume height | Radar | IMO | Volcano | Triggered by backscatt. power | NRT |
| Thermal images | TIR camera network | INGV | Volcano | Threshold nb. thermal pixels | NRT |
| SO$_2$ profiles | UV spectro. network | INGV | Volcano | Threshold SO$_2$ | NRT |
| SO$_2$ column amount | DOAS | IMO | Volcano | Threshold SO$_2$ | NRT |
| Plume height | Web camera | INGV | Volcano | Intensity contrast pixels | NRT |
| Plume height | Web camera | IMO | Volcano | Intensity contrast pixels | NRT |
| Ash index | AQUA / AIRS | ULB | Volcano | NOP required | NRT |
| SO$_2$ VCD | AQUA / AIRS | AIRES / BIRA | Volcano | Threshold SO$_2$ & coeff. | NRT |
| SO$_2$ VCD | AURA / OMI | NASA | Volcano | Threshold SO$_2$, nb. pixels | NRT |
| SO$_2$ VCD | MetOp-A&B / GOME-2 | DLR | Volcano | Threshold SO$_2$, nb. pixels | NRT |
| AOD (dusts) | MetOp-A&B / IASI | ULB | Dust | Threshold | NRT |
| Ash index | MetOp-A&B / IASI | ULB | Volcano | NOP required | NRT |
| SO$_2$ BT index | MetOp-A&B / IASI | ULB | Volcano | Threshold BT | NRT |
| SO$_2$ VCD | MetOp-A&B / IASI | ULB | Volcano | Trigerred by BT | NRT |
| SO2 plume height | MetOp-A&B / IASI | ULB | Volcano | Trigerred by BT | NRT |
| Ash mask | MSG / SEVIRI | DLR | Volcano | Threshold | NRT |
| Ash column load | MSG / SEVIRI | DLR | Volcano | Trigerred by Ash mask | NRT |
| Ash top height | MSG / SEVIRI | DLR | Volcano | Trigerred by Ash mask | NRT |
| SO$_2$ VCD | S5P / TROPOMI | BIRA / DLR | Volcano | Threshold SO$_2$, nb. pixels | NRT |
| SO$_2$ VCD | Suomi-NPP / OMPS | NASA | Volcano | Threshold SO$_2$, nb. pixels | NRT |
| Aerosol index | Sentinel-3A&B / SLSTR | FMI | Volcano / Dust | Threshold Ash index | NRT |
| Aerosol top height | Sentinel-3A&B / SLSTR | FMI | Volcano / Dust | Trigerred by Ash index | NRT |
| Thermal anomaly | Terra & Aqua / MODIS | NASA - FIRMS | Fire | Relay (through FIRMS) | NRT |
| Thermal anomaly | Suomi-NPP / VIIRS | NASA - FIRMS | Fire | Relay (through FIRMS) | NRT |
| Seismicity | SIL seismic network | IMO | Volcano | Relay (through VONA) | NRT |
| Volcanic tremor | Seismic stations | INGV | Volcano | Threshold RMS amplitude | NRT |
| Gamma radiation | Network of detectors | EURDEP / ZAMG | Nuclear | Relay (through EURDEP) | NRT |

*Number of medium or high LOC (Level of Confidence) pixels in the area in the threshold radius

The implementation of alert notifications and data products requires a two steps approach. First, a specific establishment of warning criteria for the different sensors (satellite, ground-based and in situ) and for the different types of alerts (i.e. issued from the detection of volcanic, sand/dust storms, smoke from fire and radionuclide

plumes) is required. A particular attention is given to the avoidance of false notifications (e.g. due to noise or retrieval failures) or overly frequent/redundant notifications (caused by highly dispersed plumes). As a baseline, the ash/SO2 alerts criteria of some of the satellite products used in SACS (see Brenot et al. 2014), and considered by EUNADICS EWS, are summarised in Tab. 5. This includes the name of the quantity products, the type of instruments and the platform for satellite sensors, the criteria of alerts and the limitation, and the availability and

access of data (Off-line or NRT). The other products and criteria used by EUNADICS EWS are also listed in Tab.





5. The second step is to combine the information from all the products in one multi-sensors system. The EWS relies on pre-defined geographical regions, and notifies the start of an event to parties of interest (volcanic and radionuclides clouds only) as soon as a new airborne hazard plume is detected. If within a period of 24 hours, a plume is detected again in the same region (for the same quantity product) no new notification is generated (to

avoid sending redundant information). An illustration of this two steps approach of EUNADICS multi-sensors EWS is presented in Fig. 21.

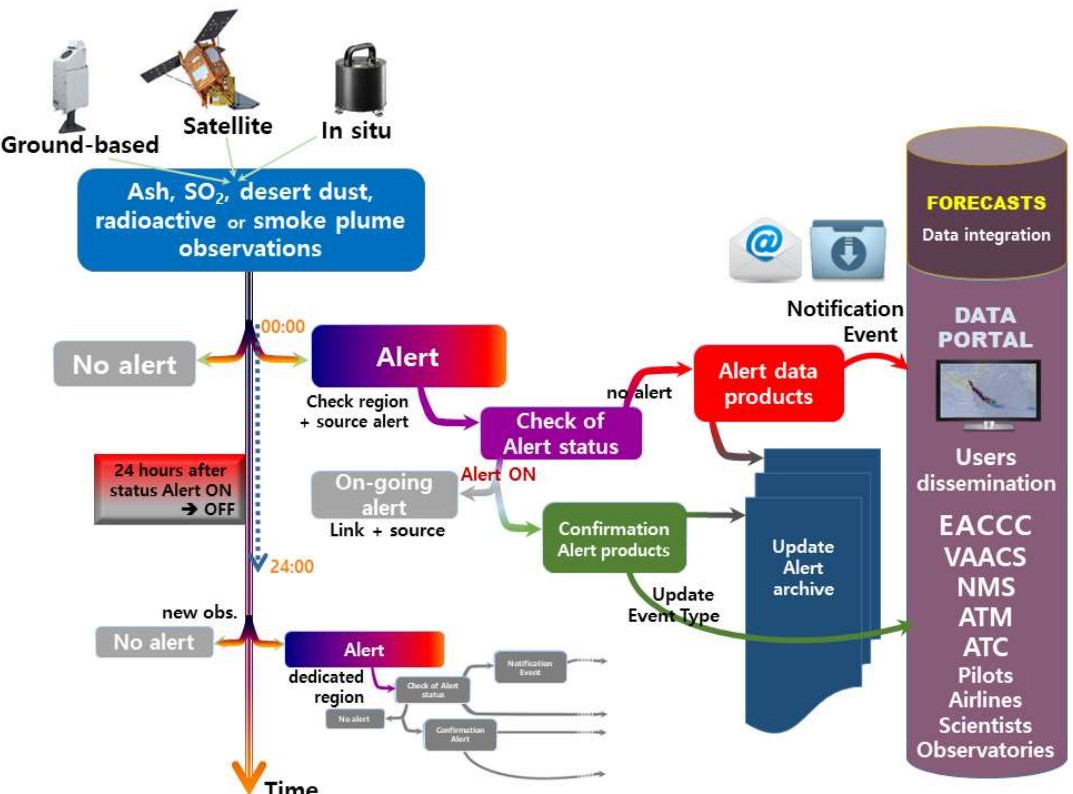

Figure 21: Illustration of the EUNADICS EWS mechanisms.

The description of the successive processes, related the first (definition of alert) and the second step (combination

of all data and avoidance of redundant information), is the following:

1) The first step is the analysis of the data using the alert criteria defined in Tab. 5. This takes place as soon as the EWS harvests new observations. After the detection of airborne emission (from a natural hazard or a nuclear source) and the start of an event, there are potentially multiple warnings generated by the system. For this reason, it has been decided to consider a

set of world regions of 30° by 30° plus two polar regions poleward of 75° in latitude. Each region is associated to a name/number for a total of 62 regions (see Fig. 9 of Brenot et al., 2014). As soon as a notification is issued, the related region is flagged 'ON'.


2) The second step is to check the "warning status" of this region. If there is no on-going notification for this region (meaning no notification since 24 hours), the warning status can
705             possibly become ON. The system compares the time of observations with the processing time. If the delay is less than 8 hours, a notification is issued. Then, alert data products are created and archived. If the "warning status" is already ON (i.e. on-going event determined using existing pool of active alert pixels; see section 4.1.4), there is no notification issued, but an update of the event is operated in the alert data archive.

This set-up enables to provide timely information to the users and also to avoid issuing too many notifications (maximum one notification per region and per 24 hours).

When an alert is issued by the system, the first step is to check whether this represents a new event (Start of Natural airborne Hazard – SNH) or it is linked (LNK) to an on-going event. At this stage, a characterisation of the source of the hazard can be obtained a) directly from the alerting process (e.g., the name and the height of the
plume of a volcanic eruption, as provided by a VONA message, or the name and location of nuclear central facing an incident), b) or by the "mother alert". In fact, if the new alert is linked to a previous event (i.e., proximity of the new alert pixels with previous alert pixels of an on-going event), the source of the hazard can be transferred. To obtain the LNK status, a proximity, in space and time (between new alert pixels and previous ones), is required to link two alerts together; see the description alert products in section 4. The proximity setting is specific to each
type of event (volcano, dust, smoke or nuclear) and the observational technique of each instrument. Generally, the distance criteria range between 500 km to 1000 km (respectively for a time-threshold of 14h or 26h). This depends on the mean revisiting time of each region, i.e., the frequency of observations for the same region. Note that sometimes the source cannot be determined and is considered as "unknown".

EUNADICS-AV EWS is based on the detection of volcanic ash/$SO_2$, sand/dust storms, smoke plumes and nuclear
accidents. If new alert pixels drive the start of a new event (SNH status), our EWS creates a log file and an internal specific notification. This triggers the sending of an email to stakeholders or public users (currently only in the case of volcanic hazard), or simply the sending of an email to the management of EUNADICS EWS (prototype status/check). A notification is associated to the creation of a dedicated webpage related to the event. If a new alert is linked to an on-going event, a confirmation of alert is established with creations of new mass, alert products
and an update the alert webpage (see section 4.6.2). A new detection of alert pixels (status SNH or LNK) is associated to the implementation of an event, which includes the reference number (date of the first alert) associated to alert products and links to quicklooks from EARLINET, E-PROFILE or other observations from the volcanic observatories. The collection of NetCDF alert data products (NCAP) and the associated data directory of an event type (ASH, SO2, DST, SMK or NUC, respectively for ash, $SO_2$, dust, smoke and radionuclide clouds)
is created, with the objective of triggering dispersion model. Access to EUNADICS partners and key users is assured via ftp or https connection.


### 4.6 Alert products

For each alert/event issued by EUNADICS EWS, the associated event type is created/updated, and the alert archive is completed. Figure 22 shows the three kinds of alerting production related to an event.


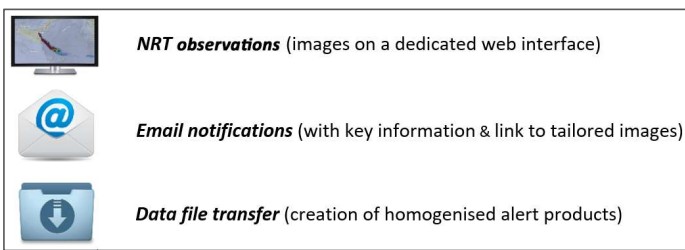

**NRT observations** (images on a dedicated web interface)

**Email notifications** (with key information & link to tailored images)

**Data file transfer** (creation of homogenised alert products)

**Figure 22: Overview of the three types of alert products developed by EUNADICS EWS.**

The diagram in Fig. 1 shows the links between the blocs of activities of EUNADICS and the red arrows characterise the specific EWS processing chain (see also Fig. 20), from observation retrievals, data harvesting,

triggering with the implementation of alerts, to the delivery of alert data products for modelling integration and ATM stakeholders.

### 4.6.1 NRT observations

NRT observations (from satellite, ground-based and in situ platforms/instruments are provided by EUNADICS partners, external data sources or thanks to agreements; see Tab. 2 and 3). A prototype data portal has been

implemented in a demonstration phase of EUNADICS project (see the exercise study of Hirtl et al., 2020a). In case of a future operationalisation of EUNADICS activity (TRL higher than 5), all the NRT observations will be visible on the EUNADICS data portal.

The Routine data products, based on NRT products from 8 satellite hyperspectral sensors (i.e. OMI, OMPS, GOME-2B, GOME-2C, TROPOMI in the UV-vis, and AIRS, IASI-A, IASI-B in the IR range) related to the

detection of volcanic eruptions, sandstorms or smoke from wildfires, and can be consulted and monitored through the SACS/EUNADICS web interface. The currently operational SACS website (https://sacs.aeronomie.be) is a self-contained system that allows the consultation of NRT satellite data and provided alerts to subscribed users in case of detection of elevated amounts or concentration of volcanic emissions. Within EUNADICS project, the development of a new SACS interface has started. This work in progress is based on modern visualisation methods

and handling of geophysical data. It is currently in development phase and allows to monitor user-selected satellite sensors and products in the form of zoomable/pannable maps, using GeoTiff, GeoServer and Web Map Service (WMS) facilities for serving geographical data. At the moment, all the NRT observations linked to EUNADICS EWS relies on the current SACS web interface. Figure 23 shows NRT observations of a volcanic burst from a paroxysm at Etna on 28 February 2021). Note that currently, the NRT images from GOME-2 correspond to

GOME-2B images for $SO_2$ and cloud cover, and to a composite images of GOME-2B and GOME-2C for the Aerosol Absorbing Index images (AAI).






**Figure 23: NRT observations on 28 Feb. 2021 during a paroxysm of Etna. Snapshot from SACS web interface (https://sacs.aeronomie.be/nrt/index.php?Year=2021&Month=02&Day=28).**


### 4.6.2 Email notifications

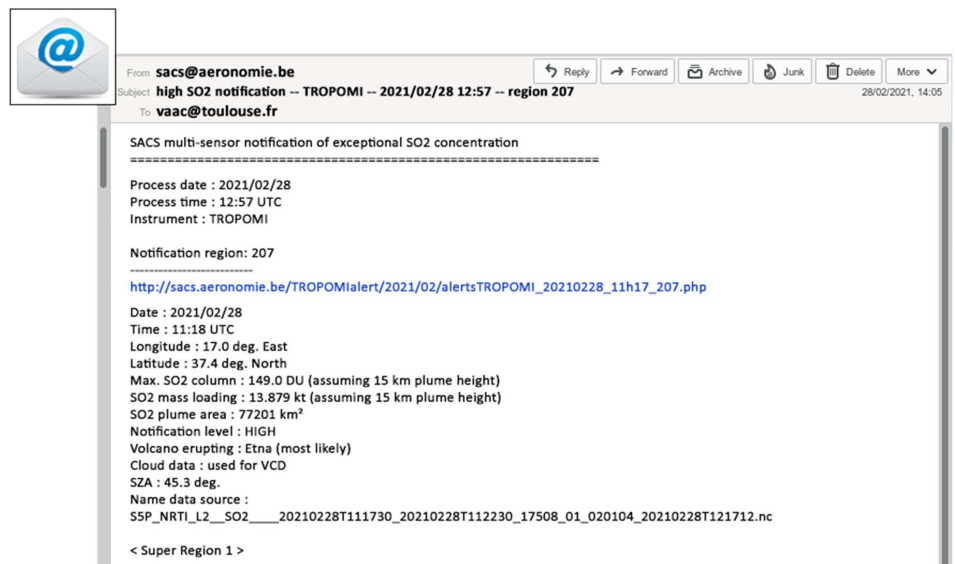

**Figure 24: TROPOMI notification after a paroxysm from Etna on 28 February 2021. A link to an alert webpage alert is provided (http://sacs.aeronomie.be/TROPOMIalert/2021/02/alertsTROPOMI_20210228_11h17_207.php).**

Currently email notifications from EUNADICS EWS for public or governmental users take only place after volcanic and radionuclide cloud hazards detection. Data products are collected by EUNADICS data harvesting facility and transferred in NRT for analysis by our EWS. The automated EWS applies specific mechanisms to issue selective detection (extension of the SACS system to other kinds of alerts and instruments) but also take into account inputs from existing systems, like NASA-FIRMS, VONA messages and EURDEP/EMERCON messages. In case of the detection of a natural airborne hazard in a specific SACS region (see section 4.5), a notification is sent internally to EUNADICS partners (one notification per affected region). In case of exceptional $SO_2$/ash concentration detected, a notification is sent to stakeholders (email) with relevant information (e.g., time, position and highest value detected) and a link to a dedicated webpage (see Fig. 24). The SACS system has currently about 300 users (from the VAACs, NMS, scientific institutions, airlines, pilots, other ATM institutions, and other public users).

On this webpage, images of volcanic observations (e.g., ash, $SO_2$ vertical column, $SO_2$ layer height if available for the instrument in alert) are shown. Additional links to other images are provided (i.e., links to interpolated plot and google earth file), as illustrated in Fig. 25 after a paroxysm at Etna on 28 February 2021. This TROPOMI $SO_2$ notification contains information about the time of observations, the lat/lon position and the value (in DU) of the highest alert pixel detected (in the dedicated SACS region), the $SO_2$ mass loading (in kt) of the data granule used to issue the alert, the size of $SO_2$ plume area (in km²), the notification level (LOW or HIGH), the name of the most likely volcano source of emission (if identified), information of the use of cloud data in $SO_2$ vertical column density (VCD), the Solar Zenith Angle (SZA in deg.; high SZA values are filtered by our system as they can show artefact $SO_2$ VCD values), and the name of the alert data product ($SO_2$ NCAP) created by EUNADICS EWS (see next section 4.6.3). More details about the notification information can be find in sections 4.6.3, in particular for the definition of completed plume considered, the notification level and the characterisation of the





source. Note that if two SACS regions are affected by an eruption, the same SO$_2$ mass can be provided in two successive SO$_2$ notifications related to 2 different regions (however the max values are different). Figure 25 shows an illustration of the tailored visualisation (of an SO$_2$ alert from TROPOMI) available from the alert webpage created by EUNADICS EWS. The link to this webpage is provided in the email SO$_2$ notification (Fig. 24).

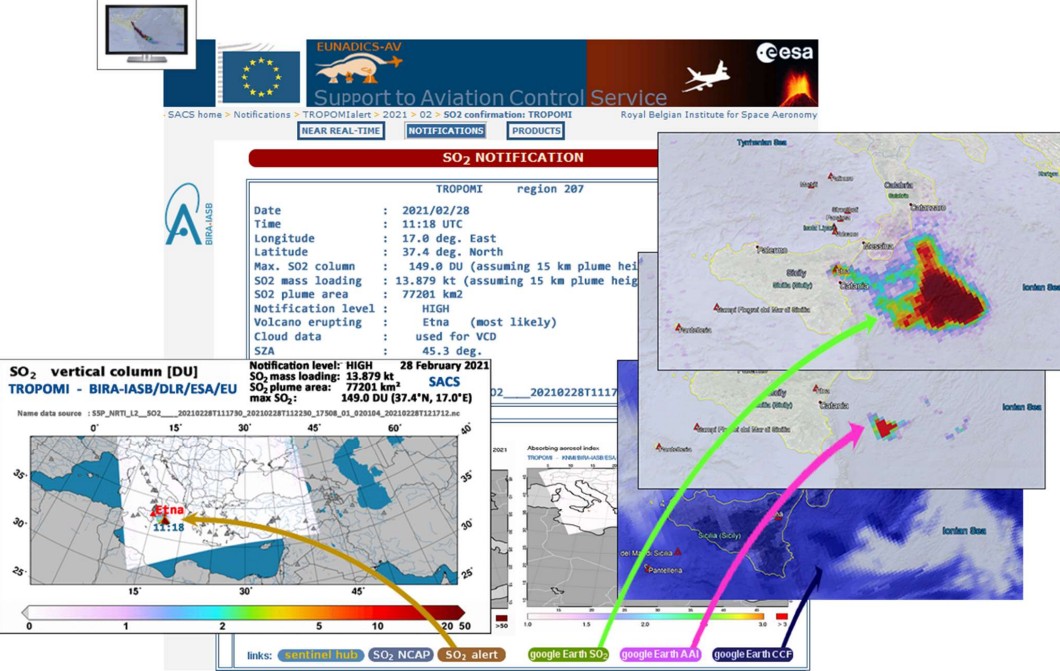


**Figure 25: Example of alert webpage with illustration of visualisations offer for a paroxysm at Etna on 28 February 2021, i.e. notification information, SO$_2$ alert image, and SO$_2$, Aerosol Absorning Index (AAI) and Cloud Cover Fraction (CCF) © Google Earth images.**

Notification information is also shown on this page. A link to a tailored SO$_2$ alert image is provided. This image
contains key information (in the legend), and shows the SO$_2$ images with the time of alert (near the max value) and the name/position of the identified volcano source (shown by a red triangle). There links to Google Earth images (SO$_2$, AAI and CCF) are indicated. We can see precisely the SO$_2$ plume, the possible ash aerosol and the cloud cover situation. This last information about the cloud cover is essential to know if the SO$_2$ observation is optimal. Indeed, the vertical SO$_2$ columns measured by UV-vis satellite sounders only considers the SO$_2$ amount
above the cloud and underestimate the possible concentration inside or under the cloud (see the limitations presented by Brenot et al., 2014). It is also important to mention the role of the cloud in the AAI estimates. Kooreman et al. (2020 revealed that several structural features can be distinguished in the TROPOMI AAI (i.e. cloud bow, viewing zenith angle dependence, sunglint, and unexplained increase in AAI values at extreme viewing and solar geometries). In the case study shown in Fig. 25, the high AAI values are not due to the sunglit
effect. We can see that the delimitation of the AAI cloud correspond precisely to the occurrence of high CCF values (Google earth facility is quite convenient for this kind of investigation). This leads us to think that the ash cloud (observed by AAI in clear sky condition and highlighted in Fig. 25) is probably not observed under the cloud/aerosol structure (area shown with high CCF). The AAI pattern observed here reduces probably the real



size of the ash cloud at low altitude (under the cloud cover). Note that the sunglit effect is systematically flagged
(and avoided) in the NRT TROPOMI AAI images available from SACS website. However, the AAI Google Earth
images do not use this flag. This is a way to see the whole scene (eventually with artefact AAI high values, but
also eventually with natural airborne aerosol observations dismissed by the sunglint flag).

If a nuclear accident takes place, EUNADICS EWS send a notification to authorities (restricted dissemination of
the information). An example of EMERCON message is shown in section 4.3 (Fig. 18).

### 830 4.6.3 Data file transfer

The data integration in dispersion models is essential in the resilience process and the decision-making after a
crisis in aviation related to airborne hazard. It is also critical for the ATM stakeholders to receive homogenised
and easy-to-read data to have a fast and clear view of the scene during such a crisis (Bolić and Sivčev, 2011,
2012). This is why one of EUNADICS objectives was the implementation of alert data products (with metadata,
key information about the alert, flag and gridded data), allowing a good dissemination of information.

The data file transfer established by EUNADICS EWS consists in the creation of Alert Products in a homogenised,
standardised, format (NetCDF), so called NCAP file. The NetCDF format has been chosen because this is a very
common and convenient format (easy access), with relevant metadata information for users. Routine alert product
is operated using a Python script that handle data collection, alerting, and database maintenance functions.


The content of the alert products is the following:

- Data (all pixels from satellite, ground-based or in-situ products )
- Tailored products:
  - alert pixels (position, values / columns densities / index, height)
  - level of severity (LOW, HIGH)
  - extended plume of hazard (completion of the hazardous plume)
  - surface and mass loading
  - gridded data
  - contours (surface, mean, max, mass)
  - information about the source, SACS region, max values
  - traceability and tracking of event (START → END)
  - links (SACS images, SACS notification, quicklooks from EU or regional network)

More details about the description of the content of the NCAP files is presented in Appendix 1.

Figure 26 illustrates an example of NCAP (dust from OMPS). An snapshot of an overview of a NCAP fiel using
hdfviewer tool is shown. Arrows in black show associated the data field with images (all pixels, dust alerts pixels,
extended dust could, and the contours identified with mean values). See section 4.1.4 for more detailed about this
triggered detection of dust using OMPS AAI product. The inventory of the NCAP implementation in EUNADICS
EWS is presented in Table 6.



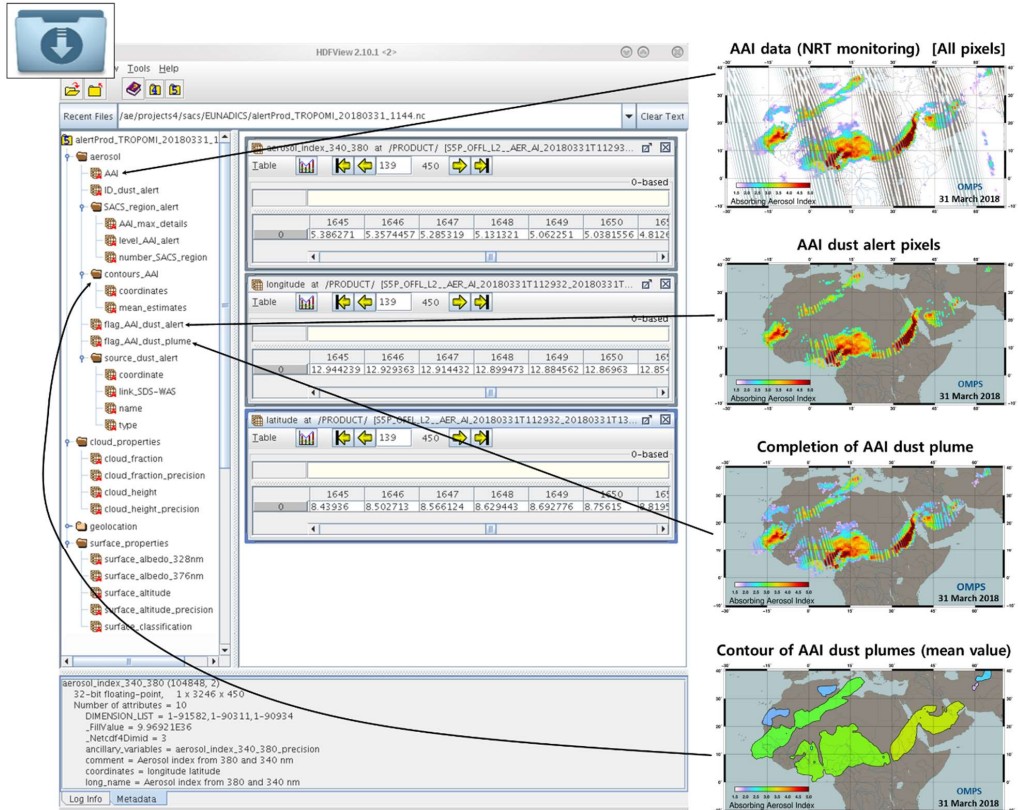

**Figure 26: Illustration about the content of a NetCDF Alert Product (NCAP) file related to the Saharan sandstorm detection from OMPS on 31 March 2018.**

865          **Table 6: Inventory of selective detection and alerting products (NCAP).**

| Platform (Satellite / Ground-based / In-situ) | Instrument | Observation | Type of detection | Source alert | Time delivery / Time resolution |
|---|---|---|---|---|---|
| Sat.: MetOp-A & -B | IASI | SO₂ / SO₂ height / Ash index | Selective | Volcano | 2h / 2 (20) per day |
| Sat.: Aqua | AIRS | SO₂ / Ash index | Selective | Volcano | 3h / 2 (20) per day |
| Sat.: MetOp-A & -B | GOME-2 | SO₂ | Selective | Volcano | 2h / 1 (10) per day |
| Sat.: Aura | OMI | SO₂ | Selective | Volcano | 3h / 1 (10) per day |
| Sat.: Suomi-NPP | OMPS | SO₂ | Selective | Volcano | 3h / 1 (10) per day |
| Sat.: Sentinel 5p | TROPOMI | SO₂ | Selective | Volcano | 2h / 1 (10) per day |
| Sat.: Sentinel 3-A & -B | SLSTR | Aerosol index / Aerosol height | Selective | Volcano & Dust | 2h / 1 (10) per day |
| Sat.: MSG-10 | SEVIRI *(geostationary)* | Ash / Ash height *(over EU only)* | Selective | Volcano | 45min / every 15min |
| Sat.: MetOp-A & -B & -C | IASI | Aerosol optical depth | Selective | Dust | 12h / 2 daily |
| Sat.: Terra & Aqua / Sat.: Suomi-NPP | MODIS } VIIRS | Fire radiative power *(NASA/FIRMS)* | Selective | Smoke | 3h / 4 (40) per day |
| Sat.: Suomi-NPP | OMPS | Aerosol index | Triggered | Dust & Smoke | 3h / 1 (10) per day |
| Sat.: Terra & Aqua | MODIS | Aerosol optical depth | Triggered | Smoke | 3h / 1 (10) per day |
| GB: EARLINET | Network Lidar | Vol. depolarisation ratio | Selective / Link quicklook | Volcano & Dust | 1h / every 30s |
| GB: E-PROFILE | Network auto. Lidar & Ceilo. | Wind prof. / Att. backs. coeff. | Link quicklook | Volcano & Dust & Smoke | 15min / every 1min |
| GB: IMO | Weather Radar | Reflectivity / height plume | Link quicklook | Volcano | 1h / every 5min |
| In-situ: VONA (IMO, INGV) | Sismo. / Camera / others | Message (obs.) | Selective | Volcano | Few minute / crisis |
| In-situ: EURDEP | Network sensors | Gamma radiation | Selective | Nuclear | 15min / every 1min |

The public access to NCAP file is currently possibly via a link in SACs notifications (related to volcanic activity only). For

https://sacs.aeronomie.be/alert/TropomiNrt/2021/02.alt/28/img/SO2_202102241052_202102281322_LNK_TR OPOMI.nc is the link provided in the alert webpage (Fig. 25) of the notification (Fig. 24) result of the paroxysm at Etna on 28 February 2021. The access to other EUNADICS NCAP data based is currently restricted to key stakeholders, i.e. VAACs, MWOs, volcanic observatories, and research collaborators.

**5 Performance verification, conclusion and future developments**

Within the system definition and design of EUNADICS EWS, a review of system requirements has been considered (see section 2). As part of our EWS, an assessment of the NRT capability of the system has been undertaken for all the products implemented in EUNADICS system (monitoring and alerting production from satellite and ground-based instruments related to the detection and situational awareness of natural, i.e. sandstorms, volcanic eruption and wildfires). The performance of the alerting approach developed within EWS is

illustrated in Fig. 27 (Month of July 2019). For four months in 2019 (May to August; only July is shown; see additional materials for the 3 other months), the selective detections of airborne hazards are schematically depicted in the form of maps (one for each month). Each detection is represented by a colour-coded dot (smoke, dust, ash and SO$_2$). A number of events can easily be identified: the eruptions of Raikoke (Russia), Ulawun (Papua New Guinea), Ubinas (Peru), wildfires in Canada, Amazonia and Russia, desert dust plumes (every months).

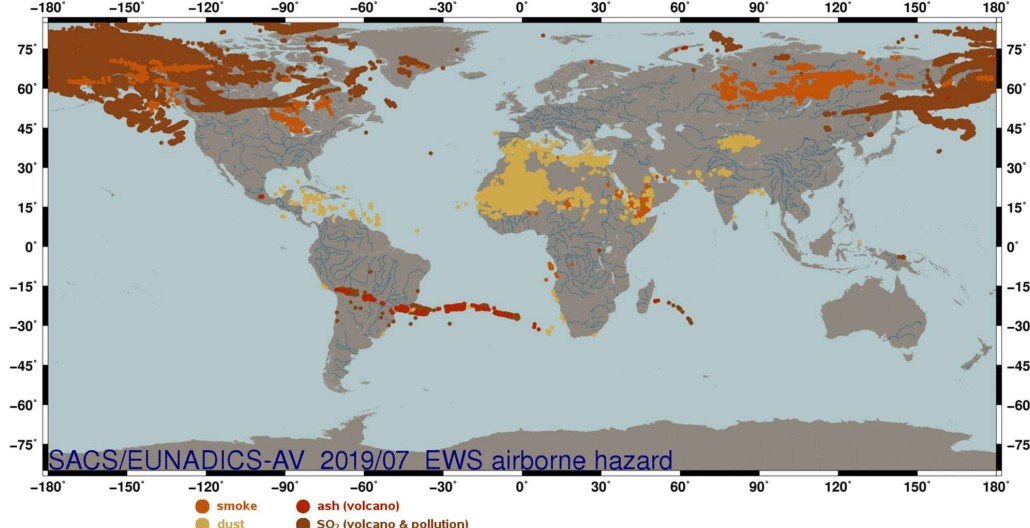


**Figure 27: Selective detections of natural airborne hazards from EUNADICS EWS in July 2019 (Raikoke and Ubinas volcanic eruption, dust storms in Sahara and Gobi deserts, and forest fires in Russia and Canada).**

EUNADICS is a SESAR (Single European Sky ATM Research; https://www.sesarju.eu) enabling project with regard to the definitions provided in the SESAR 2020 Programme Execution Framework, delivering SESAR

Technological Solutions. EUNADICS has developed, verified and validated products, and its EWS has be designed in function of potential future deployment. EUNADICS pass maturity phase V2 with regard to the 7-phase concept as introduced by the European Operational Concept Validation Methodology (E-OCVM, 2010),



which shows the feasibility of EUNADICS prototype service. EUNADICS EWS has developed a concept for starting V3 validation (i.e. pre-industrial development & integration). With regard to the alert products developed, a verification of requirements has be performed and a verification of EUNADICS EWS performance obtained (Fig. 27). A validation of the global concept of EUNADICS and its potential benefits, has been demonstrated during the EUNADICS exercise (Hirtl et al., 2020a), showing the benefits in a limited framework.

The development achieved in EUNADICS EWS shows the significant interest in using selective detection of natural airborne hazards from polar orbiting satellite. The combination of several sensors inside a single global system demonstrates the advantage of using a triggered approach to obtain selective detection from observations, which cannot initially discriminate the different aerosol types. Satellite products from hyperspectral UV and IR sensors (e.g. TROPOMI, IASI and SEVIRI) and retrievals from ground-based networks (e.g. EARLINET, E-PROFILE and the regional network from volcanic observatories), are combined by our system to create tailored alert products (e.g. selective ash detection, $SO_2$ column and plume height, dust cloud and smoke from wildfires), with identification and traceability of extreme events.

To conclude, EUNADICS EWS has development new tailored alert products for aviation, i.e. NRT observations, notification and the implementation of NetCDF Alert data Products (NCAP). EUNADICS EWS achievements concern:

- the improvement of the NRT discrimination between volcanic ash and other aerosols (dust or smoke) or meteorological clouds
- the NRT retrievals of plume heights (ash and $SO_2$)
- the NRT retrievals of volcanic ash mass loadings with use of SEVIRI onboard a geostationary platform
- the use of polar orbiting NRT measurements with better spatial resolution
- the use of key measurements from the ground-based network, in particular lidars and ceilometers measurements, as well as near-source parameters from volcanic observatories

Only the aspect of EUNADICS related to our early warnings system is presented in this study. The better characterisation of the source obtained by EUNADICS EWS is complementary and beneficial for other developments of EUNADICS consortium. With a demonstration exercise, Hirlt et al. (2020a) shows the interest of EUNADICS system in the route optimisation of the European air space during a volcanic eruption of Etna. EUNADICS activity about the assimilation, forecasts and inverse modelling, and the characterisation and the impact of the source term in dispersion modelling, is presented by Hirtl et al. (2020b) and the two recently submitted studies from Plu et al. (2021a, 2021b).

EUNADICS consortium will now target an operationalisation of its activity, in the frame of SESAR H2020, with the objective of completing the TRL 6 (demonstration in a relevant environment). EUNADICS EWS passes with success the performance verification. Concerning future plans with regard to natural airborne hazard, collaborations are on-going with key stakeholders in charge of proceeding data integration in dispersion model and providing advisory for aviation (i.e. VAACs, NMS), but also in collaboration with WMO SDS-WAS, in the frame of InDUST COST action (https://cost-indust.eu) with the use of EUNADICS dust alert products. The functionality of EUNADICS EWS is currently evolving thanks to Engage-KTN catalyst funding (https://engagektn.com/cf-summaries). The bridge between EUNADICS development and SWIM (System-Wide


Information Management) has started. Indeed, the registry of OPAS (Operational Alert Product for ATM via SWIM) as Yellow Profile SWIM, i.e. as a Notification Service of volcanic $SO_2$ height, is active since January

2021 (see https://eur-registry.swim.aero/services). $SO_2$ layer height from TROPOMI is now available in SACS monitoring system, with notifications. A validation of this new product (from BIRA) is a work in progress (Hyman and Pavolonis, 2020; Corradini et al., 2020; Hedelt et al., 2019). In addition, the development of EUNADICS EWS is also used and contributes to a recent SESAR H2020 project, which has the objective to upgrade the EUNADICS prototype EWS with other hazard to aviation. In addition to natural airborne hazard (volcano, dust

and smoke), the ALARM project (multi-hAsard monitoring and earLy wARning system; https://alarm-project.eu) plans to develop early warning and NCAP files with respect to Space Weather, Severe Weather and Environmental hotpots risk to aviation. This new activity of EUNADICS/ALARM EWS might join the ARISTOTLE consortium basis (http://aristotle.ingv.it) in the future.

Concerning the operationalisation of EUNADICS with regard to nuclear accident, the European network of

experts, called Ring of Five (Ro5), will be approached also to become part of their mailing distribution system that is used whenever one of the Ro5 laboratories detect something anomalous in their measurement networks. Although in this case, unlike with the EURDEP data, the data will not be harmonised, it can be used as a triggering system or, at least, as an awareness system potentially for such events for which the gamma dose rate monitoring may not provide useful information (very far away sources or for radionuclides that are pure beta emitters, for

instance). We can clearly see the interest of EUNADICS consortium to proceed a future relay of radiological data (gamma dose rate and radionuclides concentrations in ground-level air), to create early warnings using homogenised critical dataset, to be used to trigger data assimilation / inverse modelling for source term estimate.

**Appendix 1: Description of the NCAP files**

All the data pixels provided in a NCAP file corresponds to all the data information relevant for data integration

(for example from IASI and TROPOMI satellite sensors information about the uncertainty of measurements is provided, e.g. $SO_2$ VCD for 8 altitudes from IASI and the averaging kernel from TROPOMI; see Clarisse et al., 2011, 2012, 2014, and Theys et al., 2017, 2019). For the lidar data, all the data observations correspond to the back scattered coefficient of the whole profile (the same for te alert is a flag-value along the profile with low, medium and high alert status). For the radionuclide data, it corresponds to all the gamma radiation data from

EURDEP network.

About the level of severity (LOW, HIGH), for an $SO_2$ alert the HIGH level is obtained if the mass loading is over 5 kt. For other satellite alerts, we have defined the level of severity considering the area affected by alert pixels (if this area is over 100000 km², this brings a HIGH level). For the EARLINET, the criteria is based on the number of high alert pixels. If this number is over 10, the level is HIGH. The criteria for the level of severity is an arbitrary

choice of our system, which can be easily changed if we find this is not appropriate. For example for the $SO_2$, email notification are only sent if the level of severity/notification is HIGH. Initially, we choose a threshold of 10 kt (for a plume high altitude assumed at 15 km). This threshold has been moved to 5kt. For nuclear incident, if an EMERCON message is sent, this is automatically HIGH, there is no LOW level.

The extended plume of hazard is determined by using a lower threshold applied for the neighbour pixels of an

alert (see section 4.1.4). For example, if the threshold of an $SO_2$ alert for a sensor, is 2 DU. We can extend the



plume with the neighbouring pixels with lower SO$_2$ values (e.g. up to 0.5 DU). Detailed of the plume extension is provide in the metadata.

About the information of source of the airborne hazard. For gamma radiation, this information is generally provided in the EMERCON message. For alert related to volcanic activity, VONA messages are ingested by

EUNADICS EWS, and used for determining the name of the erupting volcano, source of the detected volcanic emission. We also use information from Volcano Discovery (https://www.volcanodiscovery.com) and MIROVA systems (https://www.mirovaweb.it) to highlight the activity of world wild volcanoes. If there is no active volcano provided by VONA messages (or other system) close to a volcanic plume detected, our system tries to determine if there is a most likely candidate volcano which can be identified as the source of the emission. If the ID (identity)

of a source is successful (60% of the highest values are located close to the same volcano), our NCAP file provides all the information about this volcano, i.e. its name, latitude, longitude, elevation, type (e.g. stratovolcano), country, rock (e.g. basaltic / andesite basaltic). This information comes from the Global Volcanism Program (https://volcano.si.edu), and a link to GVP webpage of the identified volcano is provide in the NCAP file. The ID of the source is sometimes wrong (generally when the plume is far from the source), and we still need to investigate

this aspect to avoid as much as possible ID errors. The use of a constellation of several satellites is a way to avoid this problem. Indeed, the pool of active alert pixel is defined by a volcano ID and should be able to keep the good ID, even for a plume detected far from its source (what we call the traceability of an event, in time and space). For the other natural hazard, we plan to use information of the nature of the neighbouring ground using ESA-CCI Land Cover information. This is not yet considered by EUNADICS EWS.

In case of an on-going airborne hazard event affecting the European air space, EUNADICS EWS creates link to quicklook images in the NCAP files (e.g. links to E-PROFILE, with quicklooks of aerosol, cloud and wind observations from Automatic Lidars and Ceilometers; see section 4.2 for more details about the available observations from EU and regional ground-based networks).

The generic name of a NetCDF Alert product (NCAP) is the following:

XXX_YYYYMMDDHHMM_yyyymmddhhmm_ZZZ_SENSOR.nc

Here is some examples of names:

SO2_201105221029_201105221152_SNH_GOME2-A.nc

SO2_201105221029_201105231810_LNK_GOME2-A.nc

SO2_201105221029_201105292344_END_GOME2-A.nc

XXX (3 digits) refers to the parameter used for issuing alert

- SO2 for volcanic SO$_2$ (based on SO$_2$ column)
- ASH for volcanic ASH (based on ash index or column)

- DST for desert dust (based on AOD, AAI or attenuated backscattered coefficient)
- SMO for smoke from forest fire and biomass burning (base on AOD or AAI)
- NUC for gamma radiation from a radionuclide cloud (based on gamma dose rate)

YYYYMMDDHHMM (12 digits; year month day hour minute): it refers to the UTC time of the first alert (max

values) detected for an event.





yyyymmddhhmm (12 digits; year month day hour minute): it refers to the processing time (UTC) of this alert product.

ZZZ (3 digits): is the code of the natural airborne hazard of this Event type & Alert products. For instance:

- SNH for the Start of a Natural Hazard
- LNK for an alert product linked to a previous SNH alert product
- END for an alert product ending an event (file is empty; this is issued of no more alert products are linked 26 hours after the last LNK products)

SENSOR (from 3 to 11 digits) to refer to the name of sensors (or ground-based network) used for issuing the alert. For instance:

- GOME2-B
- GOME2-C
- IASI-A
- IASI-B
- IASI-C
- OMI
- OMPS
- AIRS
- TROPOMI
- SLSTR-A
- SLSTR-B
- SEVIRI
- MODIS-Aqua
- MODIS-Terra
- EARLINET
- EURDEP

**Appendix 2: Case studies and downloads of NCAP**

- Case 1: Ophelia event – period: 15-17 October 2017

http://sacs.aeronomie.be/EUNADICS/NCAP_201710_Ophelia.zip

Case 1 is an illustration of the so-called Ophelia event. During the 15-17 October 2017 period, the Hurricane Ophelia in the Atlantic Ocean entrained Saharan desert dust and smoke particles (from Portuguese fires) over large parts of Europe leading to unusual visibility conditions. This event had implications on aviation (several flights were cancelled).

Products:

| | |
|---|---|
| - IASI-A: AOD, dust flag | *(provided by ULB)* |
| - OMPS: AAI | *(provided by NASA)* |
| - VIIRS/MODIS FIRMS flag | *(provided by NASA)* |




- • Case 2: Eyjafjallajökull eruption – period: 13 May 2010

http://sacs.aeronomie.be/EUNADICS/NCAP_201005_Eyjafjallajokull.zip

Case 2 gives an example of satellite products for the 2010 Icelandic eruption of Eyjafjallajökull. Unusual meteorological conditions brought volcanic ash clouds over Europe, causing enormous disruption to air traffic across western and northern Europe. The transport of ash can be visualised using geostationary satellite products.

Product:

- - SEVIRI: ash column       *(provided by DLR)*

- • Case 3: Grímsvötn eruption – period: 23 May 2011

    http://sacs.aeronomie.be/EUNADICS/NCAP_201105_Grimsvotn.zip

Case 3 gives an example of satellite products for the 2011 Icelandic eruption of Grímsvötn with $SO_2$ and volcanic ash emitted at different height. The transport of $SO_2$ and ash can be visualised using a suite of satellite products.

Products:

- - IASI-A: $SO_2$ column, ash index       *(provided by ULB/BIRA)*
- - AIRS: $SO_2$ column, ash index       *(provided by BIRA/AIRES/ULB)*

- • Case 4: Raikoke eruption – period: 22-26 June 2019

    http://sacs.aeronomie.be/EUNADICS/NCAP_201906_Raikoke.zip

Case 4 is for the Raikoke volcano (Russia) that erupted explosively on the 21[st] of June 2019 and emitted copious
amounts of $SO_2$ and ash in the upper-troposphere lower stratosphere. Many instruments could detect $SO_2$ and ash from Raikoke.

Products:

- - IASI-B: $SO_2$ column, ash index       *(provided by ULB/BIRA)*
- - AIRS: $SO_2$ column, ash index       *(provided by BIRA/ULB)*
- - TROPOMI: $SO_2$ column       *(provided by BIRA)*
- - SLSTR-A and SLSTR-B: ash index, ash height       *(provided by FMI)*

- • Case 5: Desert dust storm – period: 21-22 March 2018

    http://sacs.aeronomie.be/EUNADICS/NCAP_201803_Heraklion.zip

Case 6 illustrates a strong desert dust storm that lead to the closure of Heraklion airport in Crete, Greece, on 22 March 2018.

Products:

- - EARLINET: EWS flag       *(provided by CNR)*
- - IASI-A: AOD, dust flag       *(provided by ULB)*
- - OMPS: AAI       *(provided by NASA)*



**Acknowledgments**

We thank Mike Pavolonis and VOLCAT team for providing detailed information about the status of their Early Warning System related to volcano. We thank Ruslan Borodin for providing an illustration of chart (radioactivity advisory based on conditional data), as provided by the Federal Environmental Emergency Response Centre (FEERC) of Russia.

**Financial support**

This work has been conducted within the framework of the EUNADICS-AV project, which received funding from the European Union's Horizon 2020 research programme for societal challenges – Smart, Green and Integrated Transport under grant agreement no. 723986.

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
