# Peer review of "EUNADICS-AV early warning system dedicated to support aviation in case of crisis from natural airborne hazard and radionuclide cloud."

_Natural Hazards and Earth System Sciences, 2021_

## Author Comment (AC1)

**Author's Response to Mariana Adam, CC1 (05 May 2021):**

In this author's response, the text (normal style) answers point by point to the comment of Mariana Adam (text in bold). The text in blue corresponds to the authors' change in the manuscript.

**CC1 from Mariana Adam:**

**I have several observations and it would be very nice if there will be some clarifications. Regarding Table 5, it will be very useful to have the numerical values of the thresholds given.**

Issuing alert can require the use of several criteria. Table 5 provides an indicator of the type of criteria used, e.g. "Threshold", "Triggered" or "Relay". With respect to "Threshold", this doesn't always mean the use of a single threshold/value. To clarify this, we added superscripts to guide the reader for finding more explanation about the criteria used. Whenever possible, thresholds have been added.

See this new Table 5 bellow

[revised manuscript text omitted]

The line of Table 5 related to alert from Attenuated Backscattered Coefficient (from E-PROFILE) has been removed (see explanation below), as quicklooks should not be considered as alert.

**Why don't you use particles extinction and backscatter coefficients from lidars (as mentioned in Table 3)? Moreover, the example from Fig. 13 uses particle backscatter coefficient.**

Fig. 13 shows 3 graphs (particle backscatter coefficient at 532 nm, particle depolarization ratio at 532 nm, and alert for aviation). The alert for aviation uses threshold criteria on the particle backscatter coefficient at 532 nm and the particle depolarization ratio at 532 nm. See Papagiannopoulos et al. (2020) for more details. Table 5 has been updated accordingly.

**Does the example given in Fig. 14 represent a hazard? I see it just as an illustration of the Eprofile capability. Please mention if you have any criteria for attenuated backscatter from which you can set a warning.**

The event shown there does not directly present a hazard, the smoke concentrations involved are far too low to cause any issues to aviation, but are nevertheless well detected by the ALC network so that this case can illustrate what can be achieved with E-PROFILE. Cases with large scale presence of high concentrations of hazardous aerosols are rare (but have very high socio-economic impact as we all know), so that the E-PROFILE network did not yet have the chance to capture such an event due to its young age.

No threshold of attenuated backscatter for issuing warnings has been defined up to now. Only quicklooks (and data) are available. Therefore we have decided to remove the line related to E-PROFILE in Table 5. For issuing alert using attenuated backscatter, a synergy with information on aerosol typing is required and would be judicious (unless the event is extremely strong). For the time being, we rather consider the E-PROFILE network as a tool for precisely determining the 4D (lat, lon, altitude, arrival and dissipation time) presence of aerosols, once their type has been identified by other means.

**What do you mean by 'Range of att. backscatter' in Table 5? To me, what is of interest is the pollution layer geometry (layer altitude and depth).**

We agree that attenuated backscatter is very well suited to track lat/lon/altitude/depth/timing of pollution. The line related to E-PROFILE and range of att. Backscatter has been removed from Table 5. We could define a range where we would for sure issue a warning, but this would suffer from large uncertainty (either a high false alarm rate or a lot of missed events depending on tuning), due to the impossibility to do aerosol typing with the single-wavelength elastic lidars. Therefore, we would argue that a high aerosol attenuated backscatter should best be used in combination with some typing information in order to issue a warning, hence some synergy in the EWS would need to be exploited. The big advantage of automatic lidars and ceilometers is that in contrast to EARLINET they are up and running 24/7 with very high timeliness and their spatial distribution is dense, the disadvantage, of course, is that they cannot do typing.

**Please mention the timeliness for EWS, i.e., when the warning will be issued after the event (hours).**

Information about timeliness for EWS is presented in Table 6 (i.e. Time delivery and resolution). It is not our goal to provide the time delay of the alert with respect to the start of the event. Our objective is to provide situational awareness of an event and alert data product. However about the time delay of observations, we provide:

- The time delivery, i.e. the time delay between the time of measurement and the NRT availability of data retrievals.

- The time resolution, i.e. the time delay between 2 consecutive observations of the same region. For the instruments onboard polar orbiting satellites, it depends on the latitude and the type of sensors, e.g. UV-vis or IR.

**I am a bit confused about Fig. 13. You mention that the alert uses mass concentration based on backscatter coefficients thresholds. According to Papagiannopoulos et al. (2020), the thresholds are for particle backscatter coefficients, based on given mass concentrations (eq. 9). Please correct and cite the reference. Please comment on uncertainty.**

Correct. The text has been updated accordingly. The particle backscatter coefficient is retrieved following Di Girolamo et al. (1999) with an overall error of no more than 50 %. For the estimation of the alert thresholds, the methodology employs the POLIPHON method (Ansmann et al. 2012) with known uncertainty of 20-30%. Uncertainties are discussed in detail in Ansmann et al. (2019) and Papagiannopoulos et al. (2020). The text of section 4.2.1 has been modified:

A tailored alerting product from the European Aerosol Research Lidar Network (EARLINET; https://www.earlinet.org) has been developed during the EUNADICS project (Papagiannopoulos et al., 2020). It has been designed for NRT EWS applications. Using a stand-alone lidar-based method for detecting airborne hazards (volcanic ash and desert dust; no discrimination), this product is based on the EARLINET Single Calculus Chain (version 5.1),  provides temporally high-resolved, calibrated attenuated backscatter and volume depolarization ratio (at 532 nm), and cloud mask. The vertical resolution is 7.5 m, and the temporal resolution is 30 s. From these calibrated data, further particle- products (particle- backscatter coefficient and particle depolarization ratio) can be retrieved that act as the basis of the tailored product. The final product (to be used by EUNADICS EWS) is the aviation alert for desert dust/volcanic ash with a three colour-codes. This alert product uses particle mass concentrations (pmc) based on backscatter coefficient thresholds. High level detection appears in red (almost certain detection of ash or dust aerosol with $pmc \geq 4$ mg/m³). Medium level in orange (4 mg/m³ > $pmc \geq 2$ mg/m³). Low level is in yellow (2 mg/m³ > $pmc \geq 0.2$ mg/m³). An example of alert from EARLINET is shown in Fig. 13. On 21-22 March 2018, the eastern Mediterranean, in particular Crete island, was under extreme effects of warm southerly winds pushing enormous amounts of Saharan dust – and hot air – northwards. The desert dust storm caused the closure of Heraklion airport (Crete, Greece) on 22 March 2018, and was, in particular, detected by the ground-based LIDAR from EARLINET.

**Please comment on plumes heights. So far, you give examples for ash top height and SO2 plume height estimated from satellites (Figs. 3 and 5). How this information corroborates with the total SO2 concentration (threshold of mass loading of 5 kt, page 38).**

At the moment, only $SO_2$ and ash plume height is tackled by the alert data products of our EWS. $SO_2$ column and height are simultaneously retrieved by IASI and provided in our alert data products. A threshold of 5 kt is used to determine the level of an $SO_2$ notification (i.e., HIGH versus LOW).

**On the other hand, why no lidar or ALC system is used to determine the plumes geometry?**

This is a good point about the interest of using lidar and ALC to determine the plume geometry. We plan to use such information and create alert data products in the future. For the time being, it needs more investigations, as mentioned previously, to include plume height information from lidar and ALC in this study. Note however that quicklooks (provided in NRT) are already good for providing situational awareness related to the plume geometry.

**Why the lidars are not used for smoke identification? There are many papers on aerosol type, mostly based on lidar ratio and extinction Angstrom exponent. Again, why is just volume depolarization ratio used? Moreover, why not particle linear depolarization ratio?**

Here, we applied the methodology introduced in Papagiannopoulos et al. (2020) that focuses on irregular-shaped particles such volcanic ash and desert dust. Their methodology is based on a single-wavelength depolarization lidar with no spectral information; thus, smoke plumes would be challenging to identify following their approach. The methodology uses particle depolarization ratio for the estimation of the EWS.

We hope this answer clarifies the point addressed by your observations.

Thank you for these comments and your interest for our study.  Best regards,

Hugues Brenot and co-authors.

---

## Author Comment (AC2)

**Authors' Response to Anonymous Referee #1, RC1 (21 Jul 2021):**

In this authors' response, the text (normal style) answers point by point to the comment of the anonymous Referee #1 (text in bold). The text in blue corresponds to the authors' change in the manuscript.

**RC1 from the anonymous Referee #1:**

**General comments**

**This study describes European Natural Airborne Disaster Information and Coordination System for Aviation (EUNADICS-AV) Early Warning System (EWS). The EUNADICS EWS greatly extends the existing Support to Aviation Control System (SACS) automatic alert system for airborne volcanic sulfur dioxide SO2 and ash to include other airborne hazards (dust, smoke and radionuclide clouds) with creation of multiple new alert products (email and web pages with NRT maps, data files) and convenient formats (NetCDF). These new data are provided by EUNADICS partners and external data sources. The EUNADICS system further combines satellite data with the European ground-based networks (lidar and passive) and regional measurements from volcanic observatories in Iceland and Sicily.**

**EUNADICS serves European users, primarily Volcanic Ash Advisory Centers (VAACs) in London and Toulouse that have operational responsibility for volcanic ash advisories and forecasts. New message formats (NetCDF alert data products) will facilitate using the alerts to initialize plume dispersion models.**

**There is room for English and punctuation improvements, which would make paper easier to read. Many sentences need re-wording and/or clarification. Specific suggestions are mentioned below.**

**I found the paper informative and suitable for publication after language and syntax improvements.**

OK thank you, this will be addressed.

**Specific comments**

**The aviation hazards satellite data sources are comprehensive, except for direct readout data for Iceland and Europe from Satellite Measurements from Polar Orbit (SAMPO) service (https://sampo.fmi.fi/products). Using SAMPO data would help reducing alert latency time and geographical coverage of the EUNADICS system.**

Thank you for highlighting this aspect. FMI is partner of EUNADICS. Note that we use Very Fast Delivery from FMI (i.e., over Europe) and GINA (i.e., over Alaska). The following text has been added in section 4.1.1:

Note that a very fast delivery of OMI and OMPS data retrievals (time delivery of ~45 min for northern region, i.e. near Europe and Alaska) is provided by the Finnish Meteorological Institute (FMI) and the Geographic Information Network of Alaska (GINA), as satellite data are received at Sodankylä, Finland, and Fairbanks, Alaska (see https://sampo.fmi.fi).

**Abbreviations should be explained when first used.**

OK, thank you. Done

**Consider removing abbreviation from the title.**

The only acronym of the title is EUNADICS, which is the name of the system. We would prefer keeping this acronym in the title. As mentioned in the technical corrections, the abbreviation is now explained in the first line of the abstract.

**Technical corrections**

**Abstract is not clear to a general reader, not familiar with the EUNAUDICS project. I suggest explanation of the abbreviation "EUNADICS" in the abstract.** OK done in the first line of the abstract.

**45 ATM – explain abbreviation** done

**47 have shown significant** OK thx

**48 satellite[s]** done

**51 e.g.[,]** done

**55 service[s]** done

**57 to proceed – consider changing this verb** we change by: the interest of implementing

**58 … highlighting the capability of operating early warnings … - consider re-wording** done:

EUNADICS EWS has also shown the need to implement a future relay of radiological data (gamma dose rate and radionuclides concentrations in ground-level air) in case of nuclear accident. This highlights the interest of operating early warnings with the use of homogenised dataset

**75 implication in meteorological processing… – clarify** done:

Due to atmospheric transport, airborne particle cloud may also travel to area several thousand kilometres apart from the source. Such airborne particles can impact atmospheric dynamics, bringing difficulties to understand meteorological process (Knippertz and Todd, 2012). It can also cause worrying implication and damage for the aviation (Casadevall…).

**80 particles** done

**81 satellite [data]** done

**84 It makes it possible as it can to provide information**     We think we can leave the text as it is:     It makes it possible as it can provide information

**94 https://meteoalarm.org**   done, thx for this updated web link

**149 The results - objectives?**   replacement done

**153 Copernicus Atmosphere [Monitoring] Service (CAMS)**   done

**165-166 … specialization [in] atmospheric transport modelling**   done

**Figure 1: SAMPO service**   We don't think we should include SAMPO service in Figure 1 (in the list of the existing service used by EUNADICS EWS) as technically we don't use it. We get data from GINA and FMI, not from SAMPO. If a better Very Fast Delivery (VFD) can be obtained than the one already provided by GINA and FMI, this is something we should consider in the future activity of EUNADICS. Note that we added a reference to the VFD from FMI and GINA, and we mentioned SAMPO in section 4.1.1.

**186 boards**   done

**195 i.e.,**   done

**207 were**   done

**217 possibility -> discussion with ?**   replacement done

**218 Tables 1 and 2  -> 2 and 3?**   Thx, done

**227 overpass**   done

**243 particulate matter (PM)**   as the abbreviation PM is not used in the manuscript, there is no need to put (PM). Particle matter has been replaced by particulate matter

**243 volcanic ash total column [number or mass density]**   done, using mass density

**245 averaging kernel**   done

**250 We reviewed …**   done

**252 products**   done

**253  section 2.2?**     The requirements for data integration is in section 2.3 and the inventories of observations in section 2.2, as mentioned in the text:

Requirements for the data integration (section 2.3) have been considered to define a list of data product candidates (Tab. 1 and 2) from inventories of satellite, in-situ and ground-based observations (section 2.2).

**276, 277 .. Observatory which operates …**   done

**281 e.g.,** done

**296 e.g.,** done, note that a coma after e.g. has been added in all the manuscript

**308 at NOAA** done

**312 MWOs – explain abbreviation** done

**316 aim at -> with the goal of supporting …** done

**317 satellites** done

**345 use ground observations** done

**404 when** done

**405 up to the lower stratosphere – why not in the middle and upper stratosphere?** Thank you, we replace lower stratosphere by upper stratosphere (even this is rare and extreme events)

**405 Eight? satellites sensors …** Thank you, done

**407 Yang et al., [2007] - OMI product has been replaced with conceptually new OMI SO2 product: Li et al., New-generation NASA Aura Ozone Monitoring Instrument (OMI) volcanic SO2 dataset: Algorithm description, initial results, and continuation with the Suomi-NPP Ozone Mapping and Profiler Suite (OMPS), Atmos. Meas. Tech., 10, 445-458, doi:10.5194/amt-10-445-2017, 2017.** Thank you, you are right, I removed Yang et al. 2007 in the reference. I also added Li in the references (it as missing)

**415 between 3 and 21 km, - why is the upper limit 21km?** The upper limit was chosen after careful examination of several eruptions, as a lot of false detections above 21 km were observed. This is actually expected as the sensitivity to altitude, which relies on $H_2O$-$SO_2$ spectral interferences decreases with altitude (as there is less and less water vapour).

**421 e.g.,** done

**423 expressed in Kelvin degree (K)** done

**432 missing reference: Virtanen et al., (2014)** done

**438 to define** done

**443 illustrates** done

**447 a fast? ash detection** done

**448 i.e.,** done, note that a coma after i.e. has been added in all the manuscript

**469 presented** done

**470 is based**  done

**487 is obtained ?**  OK done

**503 triggered**  we put triggers

**Figure 13, left map: should the white box show station Finokalia (Crete), shown on the right?**  Done, Finokalia, already shown "Fi", is now highlighted by an arrow and the name Finokalia has been added.

**549 ash advections have not been observed**  done

**555 networks**  done

**560, 561,566: e.g.,**  done

**608 ZAMG and STUK – explain abbreviations**  done

**609 ZAMG**  done

**610 remove "have been designed"**  done

**613 delete "proceeding". … is implemented?**  Done, the text now is:

Both data sources are ultimately released by STUK after filtering of the EURDEP data is implemented and collecting the EMERCON information

**643 new alert products**  done

**644 creates**  done

**667-670  repeat of 645-650**  OK thank you, these lines have deleted

**683 quantity product – just use product**  done

**715 nuclear central - plant?**  done

**749 remove "thanks to"**  done

**751 explain TRL**  done, this text has been added:

In case of a future operationalisation of EUNADICS activity for TRL (Technology Readiness Levels; see H2020, Annex G of the General Annexes) of 5 and higher, i.e., system prototype demonstration in operational environment, all the NRT observations will be visible on the EUNADICS data portal.

**753 i.e.,**  done

**757 allows consultation -> visualization?**  OK, done

**763 burst -> cloud**  done

**801 remove "same"**   done

**814 consider**   done

**839 is operated -> is implemented ?**   OK, done

**855 NCAP fiel -> file?**   OK, done

**857 details**   done

**P36 868 possible**   done

**870 link not found**   done. We don't know why the link in the pdf was corrupt. Anyway, now this should work. The link has been simplified and the text is the following:

https://sacs.aeronomie.be/alert/SO2_202102241052_202102281322_LNK_TROPOMI.nc is the link to the NCAP provided in the alert webpage (Fig. 25) of the notification (Fig. 24) result of the paroxysm at Etna on 28 February 2021.

**873 MWOs – explain**   done, abbreviation (Meteorological Watch Offices) is now already explained in section 3.1 of this manuscript

**890-891  was designed with the goal of …**   done

**891 passed**   done

**895 obtained -> has been demonstrated?**   done

**899 satellites**   done

**906 has developed**   done

**907 notifications**   done

**908 include**   done, concern has been replaced by include

**913 better spatial resolution – better than what?**      'better' has been replaced by 'high':

the use of polar orbiting NRT measurements with high spatial resolution (under 10 km)

**916 Only one aspect**   done

**919 interest -> usefulness?**   done

**920 of using EUNADICS system in**   done

**921 activity about -> utility for …**   done

**925,930 in the framework of …**   done

The following text:

> EUNADICS consortium will now target an operationalisation of its activity, in the frame of SESAR H2020, with the objective of completing TRL 6 (demonstration in a relevant environment). EUNADICS EWS passes with success the performance verification.

has been modified to

> EUNADICS consortium will now target an operationalisation of its activity with the objective of completing TRL 5 (validation in a relevant environment) and TRL 6 (demonstration in a relevant environment) in the framework of further SESAR developments. EUNADICS EWS passes with success the performance verification in a limited environment (TRL4).

**928 proceeding -> implementing**   done

**958 the alert**   done

**971 details**   done

**972 provided**   done

**991 e.g.,**   done

We hope this document answers properly to the specific comments addressed by Referee #1. We are very grateful for this review and all the technical corrections. We apply all of them and hope this will make the paper easier to read.

Thank you very much for this review.  Best regards,

Hugues Brenot and co-authors

---

## Author Comment (AC3)

**Authors' Response to Tatjana Bolic, Referee #2, RC2 (02 Aug 2021):**

In this authors' response, the text (normal style) answers point by point to the comment of the Referee #2 (text in bold). The text in blue corresponds to the authors' change in the manuscript.

**RC2 from Tatjana Bolic (Referee #2):**

**The paper describes the results of the EUNADICS AV project, which developed different natural hazards observation and notification products, with the goal to support aviation in the cases of airborne natural hazards. My expertise is in aviation, so I cannot judge the background scientific quality, even though it seems impressive to me - the number of different tools, observations and notifications.**

**I do have several comments, that would require minor text revisions:**

1. **In the abstract the authors say "All the ATM stakeholders (e.g. pilots, airlines and passengers) can access and benefit of these alert products through this free channel." I find this a bit strong as a statement. Any memeber of public can access these products, that is true, but it is unclear how they can benefit, as there is no explanation of the meaning of any of the products - one would need to be a scientist to understand what they are looking at. This is true even for graphical products where different colors are set for different concentrations (or similar), but there is no explanation what it means for layman - even for aviation stakeholder - what is red zone? Can I fly through it or not? If not, how far should I keep? All this to say that these products have greaat value for aviation, but they are still missing an important part which is the "translation" of its meaning for aviation stakeholders that are not meterologists or atmoshperic scientist (if this is a good term at all).**

   Thank you for this clarification and your advices. We totally agree with this and will do our best to apply this advice in the future. Hopefully this will help us to improve EUNADICS service. We have removed the term 'benefit' at the end of the abstract. The text is now the following:

   All the ATM stakeholders (e.g., pilots, airlines and passengers) can access these alert products through this free channel.

2. **In section 5 the authors say "EUNADICS is a SESAR (Single European Sky ATM Research; https://www.sesarju.eu) enabling project with regard to the definitions provided in the SESAR 2020 Programme Execution Framework, delivering SESAR Technological Solutions." I would strongy suggest to rephrase this sentence, as the project itslef is not even connected to SESAR, and the products developed are not "enabling" in the sense that is used in SESAR (enabling in SESAR means a technology that is a necessary building block of an ATM infrastructure - in a sense that without it, there is no new ATM infrastructure. I would suggest to rephrase into "supporting" or similar wording.** Thank you again for this clarification. The text has been modified:

EUNADICS is supporting SESAR (Single European Sky ATM Research; https://www.sesarju.eu). The development, verification and validation of products, and the EWS were designed with the goal of potential future deployment.

3. **Next, the authors say:"EUNADICS pass maturity phase V2 with regard to the 7-phase concept as introduced by the European Operational Concept Validation Methodology (E-OCVM, 2010)..." E-OCVM presents guidance for V1-V3 of the 8 phases of ATM products life-cycle. However, I don't think that EUNADICS can claim V2 maturity level according to EOCVM, as human factors, safety, business, environmental and standards cases were not performed for any of the products. The point of the cases is to assess the impact of the soluton on a wide set of matters in the ATM. These cases are requirements that need to be passed, in order for a solution/product to mature from V1 to V2 or from V2 to V3. The EUNADICS project could easily claim TRLs 2,3 or even 4, of the H2020 technology levels, but not V2 of EOCVM. mainly because the EOCVM requires the assessment of how the products can be implemented in ATM and what would the impact be, and that was not done (the various cases) in the project, nor was that the point of the project).** Thank you again for this clarification. The following text:

EUNADICS passed maturity phase V2 with regard to the 7-phase concept as introduced by the European Operational Concept Validation Methodology (E-OCVM, 2010), which shows the feasibility of EUNADICS prototype service. EUNADICS EWS has developed a concept for starting V3 validation (i.e., pre-industrial development & integration). With regard to the alert products developed, a verification of requirements has be performed and a verification of EUNADICS EWS performance has been demonstrated (Fig. 27). A validation of the global concept of EUNADICS and its potential benefits, has also been demonstrated during the EUNADICS exercise (Hirtl et al., 2020a), showing the benefits in a limited framework.

has been replaced by:

EUNADICS passed the maturity phase V1, and we can now target the maturity phase V2 with regard to the 7-phase concept as introduced by the European Operational Concept Validation Methodology (E-OCVM, 2010). Tests with human factors, safety, business, environmental and standards cases are still required for EUNADICS products to reach the maturity phase V2 and show the feasibility of EUNADICS prototype service, according to E-OCVM. EUNADICS EWS has developed some work of validation of the alert products (TRL 4), as shown in Fig. 27. A validation of the global concept of EUNADICS and its potential benefits, has also been demonstrated during the EUNADICS exercise (Hirtl et al., 2020a), showing the benefits in a limited framework.

4. **In line 925, what do you mean by "environment). EUNADICS EWS passes with success the performance verification."?** As highlighted in comment 3., the environment is the one of E-OCVM, with tests of human factors, safety, business, environmental and standards cases. The following text (line 925-927):

EUNADICS consortium will now target an operationalisation of its activity, in the frame of SESAR H2020, with the objective of completing TRL 6 (demonstration in a relevant environment). EUNADICS EWS passes with success the performance verification.

has been changed to

*EUNADICS consortium will now target an operationalisation of its activity with the objective of completing TRL 5 (validation in a relevant environment) and TRL 6 (demonstration in a relevant environment) in the framework of further SESAR developments. EUNADICS EWS passes with success the performance verification in a limited environment (TRL4).*

5. **Finally, a suggestion to authors regarding the TRL levels of their products, in aviation setting. A product can be deemed operational in aviation if intended end-users can access the information, understand it and make decisions based on the understood information. If the presented information is not understandable by the end-user (e.g. pilot, air traffic controller), the product will not be used, even if it is completely accurate, and reliable. That is the reason for having various cases in the EOCVM methodology - to make new technology not only work, but to be understood. Some of the next steps, in my opinion should be identification of the end-users, and tailoring of the product for their use. If the end-users are only national meteorological providers, VAACs and similar, then the TRL of EUNADICS products is very high, and probably close to operational. But, if the products should be shared with other, non-scientific types of end-users, there is still a lot of work to reach high TRL levels, and that work is mainly on making the information understandable to these users.**

Thanks a lot for this advice we will definitely consider in the future activity of EUNADICS

With respect to TRL, section 4.6.1 has been modified:

*In case of a future operationalisation of EUNADICS activity for TRL (Technology Readiness Levels; see H2020, Annex G of the General Annexes) of 5 and higher, i.e., system prototype validation and demonstration in operational environment, all the NRT observations will be visible on the EUNADICS data portal.*

6. **Please review the paper for English proofing. It is overall of good quality, but there are typos and some non-English phrases that make reading slightly harder.** Done, see answer to RC1

We hope this document answers properly to the specific comments addressed by Tatjana Bolic, Referee #2. We are very grateful for this review and all the advices provided. We apply all them, and will do our best to improve the quality of our service in the future. We also improved the text of the manuscript to make the paper easier to read.

Thank you very much for this review.  Best regards,

Hugues Brenot and co-authors